# ApoAvatar: Expressive Audio-Driven Avatar Generation via Refocused Audio-Pose Priors

## Abstract

Audio-driven human video generation has greatly improved lip synchronization. However, most methods still use audio mainly to control the mouth, while the relationship between speech rhythm and body motion remains weak. This often makes generated characters look unnatural. We present **ApoAvatar**, a diffusion-based framework that ties speaking style to motion dynamics. We introduce an Audio–Pose Prior Refocusing mechanism, which adjusts pose guidance based on audio intensity. Strong accents increase gesture magnitude, while quiet parts suppress unnecessary motion. We also design a frame-wise audio–video interaction module. It updates audio features using the current visual context and the refocused pose prior through a designed bidirectional cross-attention. This yields better short-term synchronization and motion coherence. The framework supports both pose-controlled and pose-free inference within one model. Extensive experiments on EMTD and HDTF show clear gains over strong baselines in lip–audio synchronization, gesture expressiveness, and overall motion naturalness.

## 1 Introduction

Audio-driven human animation aims to synthesize realistic portrait videos from a single image and speech audio, ensuring precise synchronization between the audio and both facial expressions and body movements. This task shows significant potential for applications in filmmaking, virtual avatars, and telepresence. Recent advances, particularly through the adoption of video diffusion models and VAEs, have led to remarkable progress in generating photorealistic talking heads with accurate lip synchronization (Tian et al., 2024; 2025; Cui et al., 2025; Li et al., 2024). Another advanced line of research extends these capabilities to full-body animation by incorporating video diffusion models and additional motion supervision (*e.g.*, pose), improving coverage beyond facial regions (Meng et al., 2025; Wang et al., 2025; Zhou et al., 2025; Song et al., 2025).

However, existing methods predominantly treat audio and pose as independent conditions for injection and subsequent processing, overlooking the inherent interdependence between them. Audio influences the expression of pose, driving elements like facial expressions, hand movements, and full-body gestures. In turn, pose reflects the characteristics of audio, such as rhythm, cadence, and emotion. This relationship highlights that audio guides the timing and dynamics of pose, while pose visually embodies the nuances of audio, with both expected to work collaboratively for the generation of expressive, synchronized, and engaging avatar videos.

The relationship between audio and pose, as two key conditions for avatar generation, remains underexplored and faces two major challenges. First, **audio–motion coupling is weak**. Many methods turn audio into specific tokens (Zhang et al., 2023; Wang et al., 2024a). In these models, audio mainly controls the lips. The body is instead guided by pose priors that are independent of the audio signal (Meng et al., 2025; Gan et al., 2025a). This means gesture strength does not match the speaking style. The results are characters with accurate lip sync but stiff and repetitive body movements. They often look robotic and lack natural expressiveness. Second, **the interaction between modalities is limited**. Current models rarely design explicit structures for audio–motion communication. Instead, audio, pose, and visual context are often merged by simple concatenation or a single round of attention. These signals then flow in parallel without meaningful feedback. As a result, audio does not adapt to the motion, and motion does not reflect the nuances of speech. These limitations are clearly visible in Figure 1. The two compared methods rely on the reference

Figure 1: **ApoAvatar produces rhythm-aligned gestures (*red* arrows) that rise and fall with the waveform, while maintaining stable identity and lip motion.** We compare our ApoAvatar against two baselines, namely Multi-Talk (Kong et al., 2025) (open-source) and OmniHuman (Lin et al., 2025) (closed-source). *Green* dashed boxes indicate static hand gestures that persist even as the audio energy increases. More qualitative demos are provided in the supplementary material.

image, which lacks strong motion cues. As a result, even though the speech contains rich prosodic variations, the generated body gestures remain static and fail to reflect the expressive dynamics conveyed in the audio.

To address these issues, we propose **ApoAvatar**, a diffusion-based framework that tackles these issues through two key modules: *(i) Audio-Pose Prior Refocusing*. It computes frame-level prosodic intensity to modulate pose embeddings. This makes gesture strength follow the speaking style and avoids over-conditioning. *(ii) Frame-Wise Audio–Video Interaction*. It refines audio features using both the visual context and the refocused pose prior. In this way, audio is not static but adapts to the ongoing motion. As shown in Figure 1, while prior methods often fail to achieve coherent audio–pose coupling, our approach produces rhythmically aligned and natural full-body gestures with accurate lip synchronization and stable head pose. Experiments show clear improvements in lip–audio synchronization, gesture expressiveness, and motion naturalness. Ablation studies further confirm the effectiveness of both modules. The main contributions can be summarized as follows:

- We propose the *Audio-Pose Prior Refocusing* mechanism that explicitly models prosody as a frame-level control signal to adaptively modulate pose embeddings, effectively bridging audio dynamics and motion generation.
- We design the *Frame-Wise Audio–Video Interaction* strategy in which an Audio DiT adapter refines audio using the current video context and the refocused pose prior, producing pose- and video-aware audio features that strengthen audio–motion coupling.
- We conduct extensive experiments on EMTD and HDTF datasets. The results show consistent gains over strong baselines in visual quality, lip–audio synchronization, audio–pose alignment, hand stability, and identity consistency. The ablation studies verify the effectiveness of refocusing and frame-wise audio-video interaction.

## 2 RELATED WORK

**Video Diffusion Models.** The progress of diffusion models (Rombach et al., 2022) in visual synthesis has motivated extensive research on video generation. Existing approaches can generally be grouped by their architectural foundations. Some methods extend pre-trained image diffusion models (Esser et al., 2024), often based on U-Net, by incorporating temporal layers or attention mechanisms to capture motion across frames (Guo et al., 2024; Wang et al., 2023). Other methods replace the U-Net backbone with Transformer-based diffusion networks, where videos are represented as spatiotemporal tokens processed jointly (Kong et al., 2024; Yang et al., 2024; Chen et al., 2025a; Wan et al., 2025). These models show better scalability and flexibility, enabling high-quality video generation across different resolutions and lengths. Such advances highlight the potential of diffusion models as strong visual backbones for complex video generation tasks, with applications spanning human motion synthesis, scene animation, and multimodal content creation.

**Audio-Driven Human Animation.** Audio-driven human animation aims to generate realistic talking avatars or full-body performances directly from speech signals. More recent efforts explore end-to-end architectures based on diffusion or transformer models. SadTalker (Zhang et al., 2023)

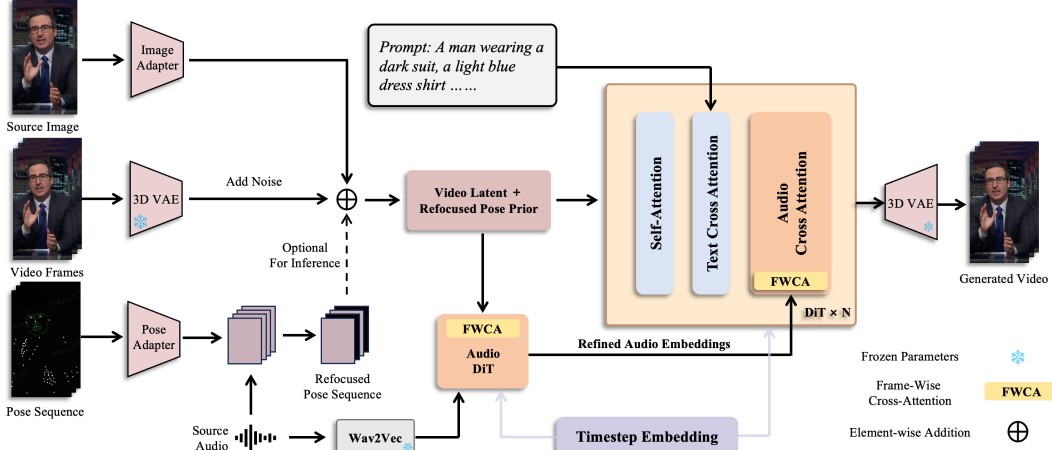

Figure 2: **Overview of ApoAvatar**. We operate in the latent space of a frozen 3D VAE. The pose sequence is encoded by a pose adapter and refocused by the audio. Processed pose features are fused into the video latents via element-wise addition. The Audio DiT adapter runs FWCA in which audio tokens query the current video latents and the refocused pose prior (with timestep conditioning), producing refined, pose-aware audio embeddings. These embeddings are then injected into the video stream via per-frame audio→video cross-attention.

and Hallo3 (Cui et al., 2025) improve audio-to-lip alignment by leveraging diffusion priors, while V-Express (Wang et al., 2024a) and EchoMimic (Chen et al., 2025c) refine naturalness by conditioning on facial landmarks or motion control signals. However, these models remain constrained to head-only generation. To address this, several studies have extended the task to larger body regions. Wan-S2V (Gao et al., 2025), CyberHost (Lin et al., 2024), and FantasyTalking (Wang et al., 2025) target half-body animation, while EMO2 (Tian et al., 2025) and EchomimicV2 (Meng et al., 2025) integrate hand pose sequences to enhance gesture realism. DisFlowEm (Sinha et al., 2025) proposes a audio-driven emotional talking head generator that disentangles pose and expression via separate optical-flow branches. OmniHuman (Lin et al., 2025) adopts a mixed data training strategy for scalable multimodal motion conditioning, and OmniAvatar (Gan et al., 2025b) further generalizes to full-body video generation. In parallel, HunyuanVideo-Avatar (Chen et al., 2025b) and MultiTalk (Kong et al., 2025) explore multi-character audio-driven animation, enabling interactive or group scenarios beyond single-speaker settings. Despite these advances, most existing methods overlook the alignment between prosodic cues in speech and body gestures, particularly hand movements.

## 3 METHOD

**Overview.** As shown in Figure 2, **ApoAvatar** generates speech-driven human videos from a single reference image $I_{\text{ref}}$, audio $c_{\text{audio}}$, text $c_{\text{text}}$, and optional pose sequence $c_{\text{pose}}$. The model runs in the latent space of a frozen 3D VAE. Section 3.1 describes the DiT backbone trained with flow matching. Section 3.2 presents Audio–Pose Prior Refocusing; it converts pose into an audio-aware prior using clip-wise prosody. Section 3.3 introduces our frame-wise audio-video interaction. An Audio DiT adapter applies Frame-Wise Cross-Attention (FWCA) between the current video tokens and the timestep embedding to obtain refined audio embeddings. Finally, Section 3.4 details our inference strategies, showing a decoupled classifier-free guidance scheme with per-modality scales.

### 3.1 PRELIMINARY

In this work, our foundational model utilizes a DiT-based video generation backbone (Chen et al., 2025a). A 3D Variational Autoencoder (VAE) is used to compress video data into a compact latent space (Yang et al., 2024). We adopt the T5 encoder (Raffel et al., 2020) to process text inputs, and the processed embeddings are integrated into the DiT backbone through cross-attention. For training, we follow a flow-matching (Lipman et al., 2023) objective, which learns a continuous velocity field to transport samples from a simple prior distribution toward the target video distribution along

deterministic paths. Specifically, the loss is formulated as:

$$\mathcal{L}_{FM}(\theta) = \mathbb{E}_{t,z_0,z_1} \left\| v_\theta(z_t, t, c) - (z_1 - z_0) \right\|_2^2, \tag{1}$$

where $z_0$ is the source distribution, $z_1$ represents the target distribution, and $c = \{c_{txt}, c_{img}, c_{audio}, c_{pose}\}$ denotes the multimodal conditions of text, image, audio, and pose (optional). The interpolated latent state is defined as $z_t = (1 - t)z_0 + tz_1$, and the model $v_\theta$ predicts its velocity at any interpolation step $t \in [0, 1]$.

## 3.2 AUDIO-POSE PRIOR REFOCUSING

To couple audio dynamics with pose conditioning, we introduce a frame-wise Audio-Guided Pose Dropout as a soft gating mechanism. Let $l_n^{(a)}$ denote the Root-Mean-Square (RMS) of the $n$-th audio frame obtained with sampling rate sr and hop length $h$; its center time is $t_n^{(a)} = n \cdot h/\text{sr}$. Let the pose stream run at $\text{fps}_{\text{pose}}$ with frame index $t \in \{0, \ldots, T-1\}$ and center time $t_t^{(p)} = t/\text{fps}_{\text{pose}}$.

We first align audio RMS to pose frames by aggregating all audio frames whose centers fall within the $t$-th pose frame window $\mathcal{W}_t = \left[ t/\text{fps}_{\text{pose}}, (t+1)/\text{fps}_{\text{pose}} \right)$. Denote the index set $\mathcal{N}(t) = \{ n \mid t_n^{(a)} \in \mathcal{W}_t \}$. We compute a per-pose loudness summary:

$$s_t = \text{LME}_\tau \left( \{ l_n^{(a)} \mid n \in \mathcal{N}(t) \} \right), \tag{2}$$

where $\text{LME}_\tau(\{x_1, \ldots, x_m\}) = \frac{1}{\tau} \log \frac{1}{m} \sum_i e^{\tau x_i}$ denotes the Log-Mean-Exp functions, aiming to softly emphasize accents (we use LME with $\tau=8$). Since perceived accents are *relative* within a clip, we apply clip-wise percentile normalization to suppress outliers:

$$\tilde{s}_t = \frac{s_t - P_{q_\ell}(s)}{P_{q_h}(s) - P_{q_\ell}(s) + \epsilon}, \quad \tilde{s}_t \in [0, 1], \tag{3}$$

where $P_{q_\ell}$ and $P_{q_h}$ are the $q_\ell$-th and $q_h$-th percentiles of $\{s_t\}_{t=0}^{T-1}$, respectively (we use $q_\ell=5$, $q_h=95$). Finally, we map normalized loudness to a pose dropout probability via *sigmoid soft gating*:

$$p_t = \sigma\left( -\alpha \left( \tilde{s}_t - \beta \right) \right), \tag{4}$$

where $\alpha$ controls the sharpness of the gate and $\beta$ sets the operating point between weak and strong audio events, and $\sigma(\cdot)$ is the sigmoid function. At training, we sample a binary mask $m_t \sim \text{Bernoulli}(1-p_t)$ and apply it to the $t$-th pose embedding. Empirically, $\alpha=10$ and $\beta=0.5$ provide a decisive yet smooth transition, effectively preserving pose signals around accents while suppressing over-conditioning during quiet segments.

## 3.3 FRAME-WISE AUDIO–VIDEO INTERACTION

We couple audio and video per frame using an *Audio DiT adapter* and an *audio injection* layer in the video blocks (see Figure 2 and Figure 3). Each aligned audio window $\mathbf{A}_t \in \mathbb{R}^{W_a \times d}$ already encodes short-term context ($W_a=5$) from the audio stream. Let $\mathbf{X}_t \in \mathbb{R}^{N_v \times d}$ be the tokens of video frame $t$ and $\tau$ be the denoising timestep.

**Audio DiT Adapter.** We modulate audio with a time-conditioned affine (Scale&Shift):

$$\widehat{\mathbf{A}}_t = \gamma_\tau^{(1)} \odot \text{LN}(\mathbf{A}_t) + \beta_\tau^{(1)}. \tag{5}$$

FWCA then refines the audio features, using the video tokens (incorporating the pose prior) of the same frame as context, with audio as queries and video tokens as keys/values:

$$\mathbf{C}_t = \text{Attn}\left( \underbrace{\widehat{\mathbf{A}}_t \mathbf{W}_q}_{\text{audio queries}}, \underbrace{\mathbf{X}_t \mathbf{W}_k}_{\text{video keys}}, \underbrace{\mathbf{X}_t \mathbf{W}_v}_{\text{video values}} \right). \tag{6}$$

We add the FWCA result back to the audio stream via a residual connection, $\mathbf{Z}_t^{(1)} = \mathbf{A}_t + \mathbf{C}_t$, and then apply a time-conditioned gate, $\tilde{\mathbf{A}}_t = s_\tau \mathbf{Z}_t^{(1)}$. The adapter runs once per denoising step and outputs refined, frame-aligned audio embeddings $\{\tilde{\mathbf{A}}_t\}_{t=1}^T$.

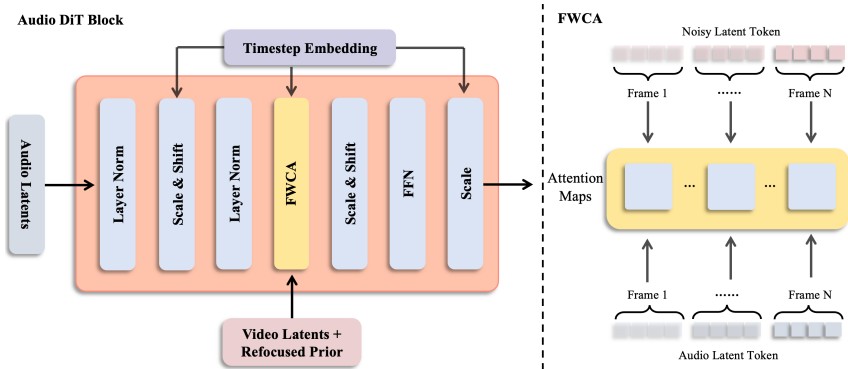

Figure 3: **Audio DiT adapter structure.** *Left:* Inside the Audio DiT block, audio latents are layer-normalized and time-conditioned via Scale & Shift from the timestep embedding. FWCA then updates the audio latents using the current video latents as context. *Right:* FWCA computes attention per frame: video tokens of frame $t$ (noisy video latents) query only the audio window aligned to frame $t$, yielding a block-diagonal attention map.

**Frame-Wise Audio Cross-Attention.** Inside each video transformer block (after self-attention and text cross-attention), we apply a per-frame audio cross-attention that mirrors the adapter but swaps the roles: video provides the queries, and refined audio provides keys/values. For frame $t$ with video tokens $\mathbf{X}_t$, the cross-attention output $\mathbf{V}_t^{\ell+1}$ is obtained by

$$\mathbf{Y}_t = \mathrm{Attn}\big(\mathbf{X}_t\mathbf{W}_q^{(v)}, \tilde{\mathbf{A}}_t\mathbf{W}_k^{(v)}, \tilde{\mathbf{A}}_t\mathbf{W}_v^{(v)}\big), \qquad \mathbf{V}_t^{\ell+1} = \mathbf{X}_t + \mathbf{Y}_t. \tag{7}$$

Attention is computed independently for each frame so that tokens of frame $t$ attend only to $\tilde{\mathbf{A}}_t$.

### 3.4 INFERENCE STRATEGIES

**Decoupled CFG.** We use classifier–free guidance with separate scales for each modality. Let $c = (c_\text{txt}, c_\text{img}, c_\text{audio}, c_\text{pose})$ and $\varnothing$ denote the null token. With guidance scales $(\lambda_\text{txt}, \lambda_\text{img}, \lambda_\text{audio}, \lambda_\text{pose})$, the guided velocity is:

$$\begin{aligned}
\hat{v}_\theta(z_t, t, c) = {}& \lambda_\text{pose}\big[v_\theta(c_\text{txt}, c_\text{img}, c_\text{audio}, c_\text{pose}) - v_\theta(c_\text{txt}, c_\text{img}, c_\text{audio}, \varnothing)\big] \\
& + \lambda_\text{audio}\big[v_\theta(c_\text{txt}, c_\text{img}, c_\text{audio}, \varnothing) - v_\theta(c_\text{txt}, c_\text{img}, \varnothing, \varnothing)\big] \\
& + \lambda_\text{img}\big[v_\theta(c_\text{txt}, c_\text{img}, \varnothing, \varnothing) - v_\theta(c_\text{txt}, \varnothing, \varnothing, \varnothing)\big] \\
& + \lambda_\text{txt}\big[v_\theta(c_\text{txt}, \varnothing, \varnothing, \varnothing) - v_\theta(\varnothing, \varnothing, \varnothing, \varnothing)\big] \\
& + v_\theta(\varnothing, \varnothing, \varnothing, \varnothing).
\end{aligned} \tag{8}$$

This gives independent, per–modality control at test time without retraining.

**Pose–Free Inference.** Because the pose branch is heavily dropped during training and refocused by audio, the model is robust when pose is absent. At test time we simply set $c_\text{pose} = \varnothing$ and $\lambda_\text{pose} = 0$ in Equation 8. The rest of the pipeline is unchanged: the Audio DiT adapter and the per–frame audio injection still provide rhythm–aware motion from audio alone.

**Pose–Conditioned Inference.** We enable the pose adapter and feed a pose sequence $c_\text{pose}$. During inference we do not sample dropout; the refined pose features are added to the video latents as in Figure 2. Audio is handled by our two–step schedule: the Audio DiT adapter produces frame–aligned $\tilde{\mathbf{A}}_t$, and each video block injects them with frame–wise cross–attention. We set $\lambda_\text{pose} > 0$ in Equation 8 to keep the pose prior active.

**Default Evaluation.** For fairness to prior methods that do not use pose inputs, unless otherwise noted we report all results in the *pose-free* setting ($c_\text{pose} = \varnothing$, $\lambda_\text{pose} = 0$).

Table 1: **Quantitative comparison of audio-driven animation methods on the HDTF dataset**. The best and second-best results on each metric are **bolded** and underlined, respectively.

| Method | Video Quality | | | | Lip Sync | | ID |
|---|---|---|---|---|---|---|---|
| | FID↓ | FVD↓ | IQA↑ | ASE↑ | SYNC↑ | SYND↓ | FSIM↑ |
| AniPortrait (Wei et al., 2024) | 96.12 | 645.72 | 1.96 | 1.15 | 7.64 | 7.79 | 0.85 |
| V-Express (Wang et al., 2024b) | 119.45 | 748.57 | 1.32 | 1.16 | 7.92 | 7.96 | 0.89 |
| EchoMimic (Chen et al., 2025c) | 167.17 | 757.38 | 1.61 | 1.19 | 6.71 | 8.23 | 0.82 |
| HyAva (Chen et al., 2025b) | 100.10 | 662.61 | 1.52 | 1.06 | 7.22 | 8.98 | 0.85 |
| Hallo3 (Cui et al., 2025) | 74.10 | **250.12** | 1.95 | 1.14 | 7.31 | 9.30 | **0.91** |
| MultiTalk (Kong et al., 2025) | 85.01 | 404.45 | 1.78 | 1.13 | 8.76 | 7.69 | 0.84 |
| OmniAvatar (Gan et al., 2025b) | 131.69 | 705.14 | 1.67 | 1.10 | **8.81** | 7.76 | 0.78 |
| **Ours** | **70.17** | 271.85 | **1.99** | **1.19** | 8.43 | **7.66** | 0.89 |

Table 2: **Quantitative comparison of audio-driven animation methods on the EMTD dataset**.

| Method | Video Quality | | | | Lip Sync | | ID | Hand Stability | |
|---|---|---|---|---|---|---|---|---|---|
| | FID↓ | FVD↓ | IQA↑ | ASE↑ | SYNC↑ | SYND↓ | FSIM↑ | HKC↑ | HKV |
| Fantasy (Wang et al., 2025) | 133.73 | 1307.20 | 2.11 | 1.12 | 1.11 | 12.88 | 0.59 | 0.57 | 8.0 |
| HyAva (Chen et al., 2025b) | 139.39 | 2160.92 | 1.76 | 1.18 | 4.89 | 9.37 | 0.67 | 0.75 | 29.2 |
| Hallo3 (Cui et al., 2025) | 104.51 | 1256.10 | 2.31 | 1.48 | 4.26 | 10.22 | 0.73 | 0.77 | 6.3 |
| MultiTalk (Kong et al., 2025) | 103.68 | 1040.43 | 2.07 | 1.30 | 6.34 | 8.47 | 0.71 | 0.79 | 14.6 |
| OmniAvatar (Gan et al., 2025b) | 82.54 | 1104.99 | 2.16 | 1.31 | 5.40 | 9.13 | 0.72 | 0.86 | 28.7 |
| **Ours** | **67.11** | **1020.54** | **2.31** | **1.65** | 6.88 | 8.17 | **0.86** | **0.93** | 14.5 |

## 4 EXPERIMENTS

### 4.1 IMPLEMENTATION DETAILS

**Datasets.** Our data processing pipeline is inspired by InfinityHuman (Li et al., 2025). First, we apply SceneDetect (Castellano, 2024) for temporal cropping of raw videos. Then, YOLO (Jocher et al., 2023) is employed to track the single person, extract corresponding spatiotemporal bounding boxes, and perform spatiotemporal cropping. We further filter videos based on multiple criteria, including overall quality, aesthetics, motion amplitude, hand visibility, mouth clarity, and the proportion of the person within the frame. This procedure results in 3,200 hours of single-person video clips used to train the pose-guided model. Building on this dataset, we additionally employ SyncNet (Chung & Zisserman, 2016) to evaluate audio–lip synchronization and select a subset of 1,500 hours of clips for training.

**Training Details.** Our audio-driven model is trained in two stages with full-parameter fine-tuning on 128 GPUs. In Stage 1, we jointly optimize the four branches (audio, pose, text, and image/video) with a learning rate of $1 \times 10^{-4}$. In Stage 2, we re-initialize the text and video branches with their original pre-trained weights, keep the warmed-up audio/pose weights from Stage 1, and fine-tune the whole model with a learning rate of $3 \times 10^{-5}$. Pose sequences are extracted with Sapiens (Khirodkar et al., 2024), and text labels are obtained by captioning the reference image using Tarsier2 (Yuan et al., 2025a) and appending a simple speaking prompt (e.g., "the person is speaking toward the camera, with body movements naturally synchronized to their speech"). We use a global batch size of 256 and generate clips at 24 fps with 720p resolution (short side) for all experiments.

### 4.2 COMPARISON WITH STATE-OF-THE-ART METHODS

**Evaluation Metrics.** For evaluation, we adopt a diverse set of metrics covering different aspects of video generation. Overall visual quality is assessed through FID (Heusel et al., 2017) for image fidelity, FVD for temporal dynamics (Unterthiner et al., 2018), and Q-Align (Wu et al., 2023) for both perceptual quality (IQA) and aesthetic preference (AES). To measure lip synchronization, we apply Sync-C and Sync-D (Chung & Zisserman, 2016). Identity preservation is evaluated using FaceSIM (Yuan et al., 2025b; Huang et al., 2024), while hand performance is examined with Hand Keypoint Confidence (HKC) and Hand Keypoint Variance (HKV).

**Test Sets.** For evaluation, we mainly rely on two publicly available datasets: EMTD (Meng et al., 2025) and HDTF (Zhang et al., 2021). The HDTF dataset focuses on talking-face scenarios, where all videos consist of single-head speech clips, making it suitable for assessing models that only drive

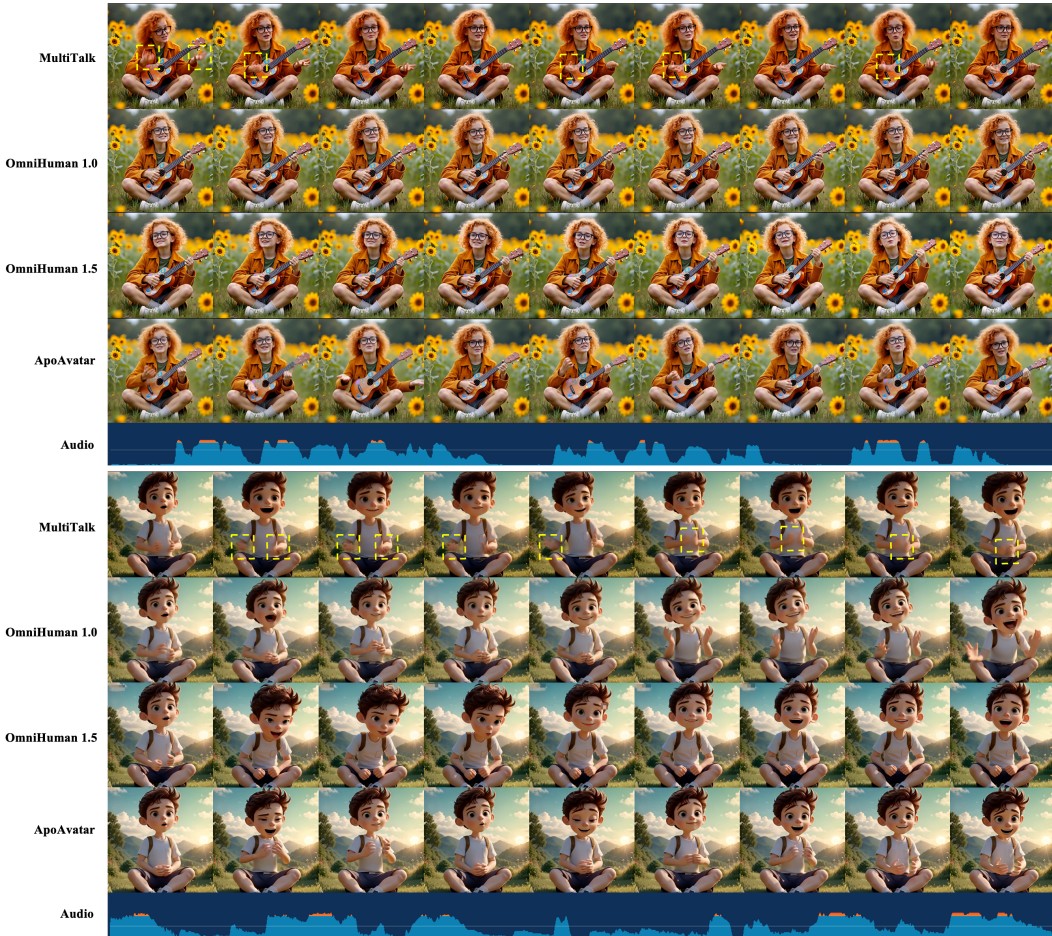

Figure 4: **Qualitative comparison with MultiTalk (Kong et al., 2025) and OmniHuman (1.0 (Lin et al., 2025)/1.5 (Jiang et al., 2025)).** For each clip we show sampled frames and the audio trace (bottom). Baselines either move a lot but yield blurry, unstable hands (yellow boxes) or stay mostly static with low expressiveness. **ApoAvatar** produces clear, rhythm-aligned gestures while keeping identity stable. Zoom in to get a better view.

facial regions. From this dataset, we randomly select 100 samples at a resolution of 512×512 for evaluation. In contrast, the EMTD dataset was originally introduced for training pose-driven models but is also well-suited for testing audio-driven portrait video generation with upper-body coverage. EMTD contains 110 videos at 720p resolution, nearly one quarter of the clips lasting more than 15 seconds. This makes it particularly useful for evaluating high-resolution audio-driven generation.

**Baselines.** We compare ApoAvatar against recent human animation methods—FantasyTalking (Wang et al., 2025), Hallo3 (Cui et al., 2025), HunyuanAvatar (Chen et al., 2025b), MultiTalk (Kong et al., 2025), and OmniAvatar (Gan et al., 2025b) on the EMTD dataset. For OmniHuman (Lin et al., 2025), we do not include it in the main quantitative evaluation because (i) it accepts at most 15-second audio inputs, which is incompatible with a subset of EMTD samples, and (ii) the model is closed-source and only available via an online demo, making large-scale evaluation time-consuming. Instead, we conduct a separate user study and a qualitative comparison on a curated set for OmniHuman. In addition, we evaluate on HDTF (Zhang et al., 2021), a face-only talking-head dataset, which enables comparison against pose-free baselines. We therefore report results against SadTalker (Zhang et al., 2023), AniPortrait (Wei et al., 2024), VExpress (Wang et al., 2024a), EchoMimic (Chen et al., 2025c), and other representative full-body methods.

**Quantitative Results.** HDTF is a classic talking head dataset. However, since it only contains upper-body segments (above the shoulders) while our model is designed for audio-driven full-body animation, evaluation on this dataset does not fully showcase our model's advantages. Nevertheless, our method still achieves remarkably competitive performance. As shown in Table 1, ApoAvatar

Table 3: **User study results of audio-driven animation methods**.

| Method | Video Quality↓ | Lip Sync↓ | ID Consistency↓ | Audio-Pose Alignment↓ |
|---|---|---|---|---|
| MultiTalk (Kong et al., 2025) | 3.48 | 3.08 | 2.85 | 3.1 |
| OmniHuman1.0 (Lin et al., 2025) | 2.04 | 2.32 | 2.33 | 2.30 |
| OmniHuman1.5 (Jiang et al., 2025) | 2.33 | 2.41 | 2.42 | 2.53 |
| **Ours** | **2.00** | **1.91** | **2.07** | **1.87** |

Table 4: **Comparison of different pose prior injection schemes on the EMTD dataset**.

| Method | Video Quality | | | | Lip Sync | | ID | Hand Stability | |
|---|---|---|---|---|---|---|---|---|---|
| | FID↓ | FVD↓ | IQA↑ | ASE↑ | SYNC↑ | SYND↓ | FSIM↑ | HKC↑ | HKV |
| Baseline (train *w/o* pose) | 117.11 | 1166.27 | 2.15 | 1.48 | 6.33 | 11.22 | 0.78 | 0.78 | 9.2 |
| + Pose Dropout | 107.61 | 1141.53 | 2.24 | 1.53 | 6.49 | 9.99 | 0.81 | 0.80 | 12.9 |
| + Temporal Pose Dropout | 99.17 | 1107.32 | 2.21 | 1.57 | 6.64 | 10.73 | 0.83 | 0.82 | 17.4 |
| **+ Refocusing (Ours)** | **67.11** | **1020.54** | **2.31** | **1.65** | **6.88** | **8.17** | **0.86** | **0.93** | 14.5 |

achieves the best FID, the best IQA, and ties for the best ASE. It also achieves the lowest SynD, indicating superior audio-lip alignment. Meanwhile, Halo3 reports the best FVD and FSIM score, while OmniAvatar yields the highest SynC. Our method remains highly competitive, ranking second in both FVD and FSIM. These results demonstrate that our model improves image realism and lip alignment while remaining competitive on identity preservation and temporal quality. On the EMTD dataset, as can be seen in Table 2, ApoAvatar delivers the best performance across all metrics. Note that the HKV statistic reflects motion frequency and is reported for reference only (with no inherent better/worse ordering). Across both datasets, ApoAvatar achieves first or second-best performance on most metrics, with clear advantages on EMTD, supporting our claims of rhythm-aligned motion, strong visual quality, and stable identity.

**Qualitative Results.** We further perform a qualitative evaluation. As shown in Figure 4, generations from OmniHuman 1.0/1.5 often keep the body largely static, resulting in limited expressiveness. While MultiTalk exhibits more movement, it frequently produces blurred or deformed hands and suffers from timing drift. This highlights a common trade-off in prior methods between motion expressiveness and temporal stability. ApoAvatar overcomes this limitation by training with a pose-aware prior. Our results demonstrate clear hand shapes, stable gestures that align with audio rhythm and emphasis, while well preserving identity and lip synchronization.

**User Study.** We conducted a large-scale preference study to assess perceptual quality. We collected 30 clips covering diverse scenes, languages, genders, poses, and video styles, and generated results from ApoAvatar, MultiTalk (Kong et al., 2025), OmniHuman-1.0 (Lin et al., 2025), and OmniHuman-1.5 Jiang et al. (2025). 80 expert raters viewed anonymized, randomized quadruplets per clip and ranked the methods on four criteria: Video Quality, Lip Sync, ID Consistency, and Audio–Pose Alignment.

As shown in Table 3, we report the average rankings for each criterion. ApoAvatar achieves the best performance across most criteria. In terms of Video Quality, our method obtains the lowest average rank (2.00), showing clear preference over all baselines. For Lip Sync, ApoAvatar obtains the best score with an average rank of 1.91, exceeding all competing methods. On ID Consistency, ApoAvatar again receives the most favorable rank (2.07), indicating better preservation of subject identity during long clips. Most notably, for Audio–Pose Alignment, our method achieves the best result (1.87), whereas other models fail to reflect prosodic variations in body motion. These results confirm that the proposed design substantially improves the naturalness and coherence of audio-driven animation, especially in aligning speech dynamics with expressive full-body gestures.

### 4.3 ABLATION STUDY AND DISCUSSION

**Ablation on Audio–Pose Prior Refocusing.** For a comprehensive abaltio analysis, we compare four variants on the EMTD dataset: (i) a backbone training without pose, (ii) training with a pose branch by a fixed pose dropout rate (drop the whole pose condition), (iii) adding *temporal pose dropout* (drop random pose frames), and (iv) our refocusing module.

Adding pose already helps over the no–pose baseline, showing that a pose prior stabilizes identity and improves motion. Switching from sample-level dropout to temporal dropout brings another step forward: the model becomes less sensitive to occasional pose errors and produces more varied

Table 5: **Comparison of different interaction schemes on the EMTD dataset**.

| Method | Video Quality | | | | Lip Sync | | ID | Hand Stability | |
| --- | --- | --- | --- | --- | --- | --- | --- | --- | --- |
| | FID↓ | FVD↓ | IQA↑ | ASE↑ | SYNC↑ | SYND↓ | FSIM↑ | HKC↑ | HKV |
| Baseline (train *w* pose) | 111.91 | 1160.58 | 2.11 | 1.34 | 6.23 | 10.13 | 0.78 | 0.84 | 16.2 |
| + frame-wise | 83.04 | 1090.57 | 2.14 | 1.48 | 6.59 | 9.17 | 0.80 | 0.87 | 11.6 |
| + modal interaction | 72.68 | 1108.77 | 2.22 | 1.44 | 6.67 | 8.96 | 0.82 | 0.88 | 20.7 |
| **frame-wise modal interaction** | **67.11** | **1020.54** | **2.31** | **1.65** | **6.88** | **8.17** | **0.86** | **0.93** | 14.5 |

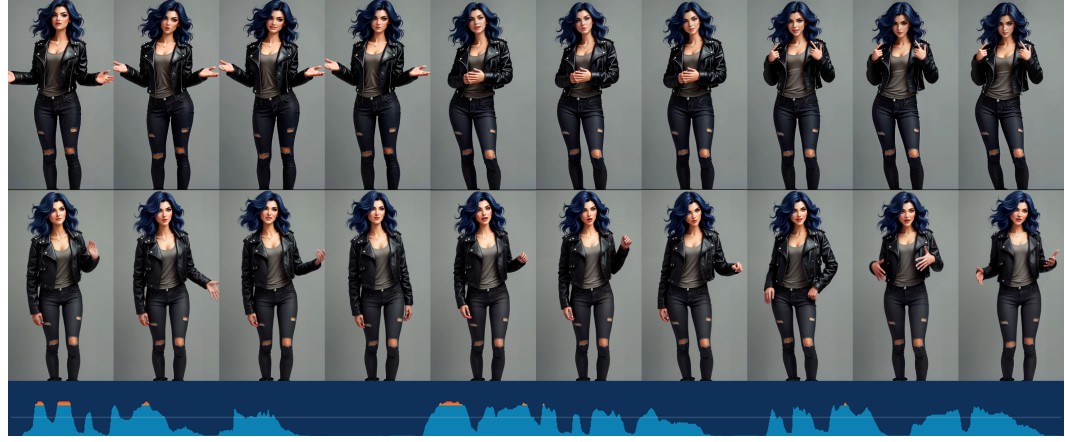

Figure 5: **Effect of Audio–Pose Prior Refocusing.** Top row: *w/o* refocusing, by randomly dropping out the pose condition in sequence, we find the model produces only simple, repetitive gestures with poor prosody alignment. Bottom row: *with* refocusing, gestures rise and fall with the audio rhythm.

hand and body movements. Our refocusing replaces fixed masking with a prosody-driven gate. Pose is retained around strong accents and softened in quiet spans, frame by frame. This change consistently improves visual quality, lip–audio alignment, identity similarity, and hand stability in Table 4. Apparently, a pose prior is useful, but making it audio-aware is key—refocusing delivers the best overall results.

Figure 5 shows a side-by-side comparison. Without refocusing, gestures are coarse, repetitive, and largely insensitive to prosody. With refocusing, motions are fine-grained and rhythm-aligned: onsets match energy increases, pauses yield natural rests, and accents trigger larger, faster upper-body movements.

**Ablation on Frame-Wise Audio-Video Interaction.** We study four variants on EMTD (Table 5). *Baseline (pose, no interaction)* uses the same Goku-I2V (Chen et al., 2025a) backbone with pose conditioning for training; audio interacts with the model through simple cross attention. + *frame-wise* adds a per-frame audio→video cross-attention in the video blocks. + *audio-video interaction* instead adds the Audio DiT adapter but keeps a clip-wise audio injection in the video stream. *Frame-wise audio-video interaction* combines both pieces: audio features are first refined by the video context frame-wise, then injected back with per-frame cross-attention in video DiT blocks.

From the baseline to + *frame-wise*, video quality and synchronization clearly improve (e.g., FID drops from 111.9 to 83.0, SYND decreases from 10.1 to 9.2), indicating that aligning audio tokens to the exact video frame is useful. + *audio-video interaction* further reduces FID and improves IQA, showing that letting audio see the current visual state helps the static appearance; however, FVD improves less than the frame-wise variant and HKV rises, suggesting more motion but not tightly timed. Our full *frame-wise audio-video interaction* achieves the best overall results: the lowest FID/FVD and SYND, the highest IQA/ASE and FSIM/HKC, and a moderate HKV, meaning gestures are both clear and well timed.

## 5  CONCLUSION

In this work, we present ApoAvatar for audio-driven human video generation that explicitly couples audio with body motion. Our key idea is Audio-Pose Prior Refocusing, a soft-gating mechanism that adaptively regulates pose embeddings by clip-wise prosodic intensity. Built on this pose-aware

prior, we introduce a frame-wise audio–video interaction strategy: refined audio features are temporally aligned to each video frame and injected via cross-attention after video self-attention and text conditioning, yielding tight lip synchronization and rhythm-consistent upper-body dynamics. The design supports both pose-controlled and pose-free inference from a single source image. Extensive experiments on EMTD and HDTF show consistent gains over strong baselines in lip–audio synchronization, gesture expressiveness, and overall motion naturalness, with ablations validating the contribution of each component.

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

# A APPENDIX

## A.1 POSE CONTROL CASES

**Inference with Pose.** At test time, ApoAvatar can run in a pose–controlled mode by feeding an external pose sequence $c_{pose}$ to the pose adapter. In this setting we bypass the refocusing gate and use the pose as a hard control (no audio-driven dropout). The model faithfully follows the given trajectory while keeping lip–audio synchronization and identity stable. Representative results are shown in Figure 6, where the hands and upper body match the input pose and the mouth motion remains natural.

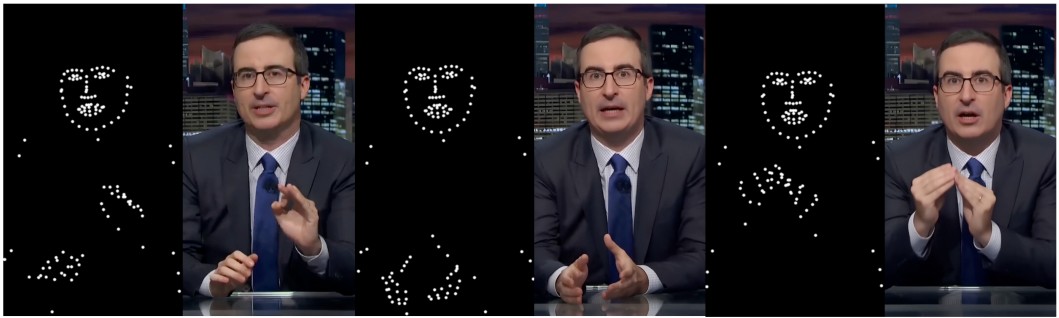

Figure 6: **Pose-driven generation.** Given a pose sequence (left of each pair), **ApoAvatar** synthesizes the corresponding frames (right). The model closely follows the driving pose and produces natural lip motion and hand gestures with smooth temporal coherence.

**Pose vs. Pose-Free Robustness.** As shown in Figure 7, explicit pose control works but is fragile. When the input contains occlusions or atypical configurations (e.g., a masked face), the extracted pose becomes unreliable and the model fails to adapt. The problem is more severe for stylized or non-human characters whose proportions differ from real humans. Our approach mitigates these issues. We use pose to build a prior during training but do not depend on it at test time. The refocusing mechanism lets audio modulate this prior and trigger gestures when prosody requires them. As a result, the pose-free mode remains stable across occlusions and character styles.

| Reference Image | Pose Condition | Pose-driven Results | Pose-free Results |
|---|---|---|---|

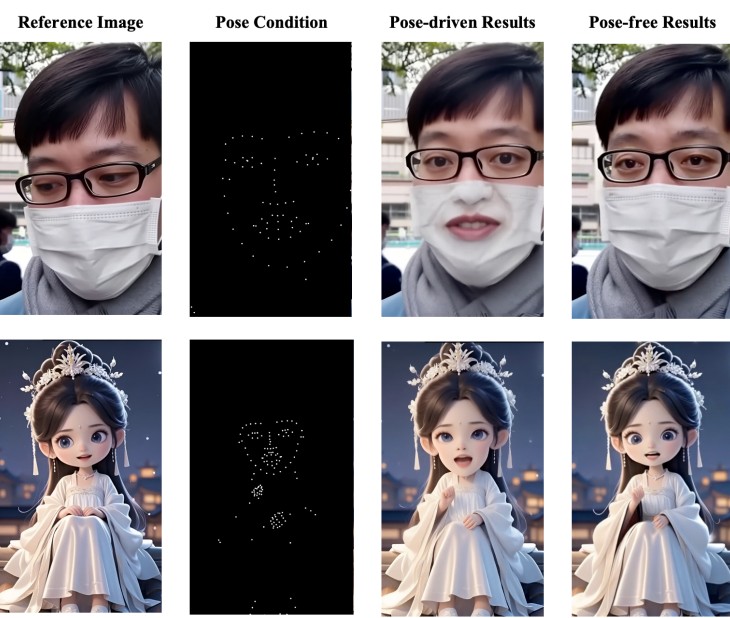

Figure 7: **Pose control can be brittle.** The pose signal can contradict the true geometry. When used to drive animation, these flawed poses yield artifacts and unnatural motion, whereas our pose-free mode remains robust, producing cleaner lips, more plausible gestures, and better identity stability.

## A.2 APPLICATIONS

**Reference-Free Generation.** Thanks to image dropout during training, **ApoAvatar** can run without a reference image. As shown in Figure 8, given only a text prompt $c_{text}$ and speech audio $c_{audio}$, the model synthesizes a plausible identity that matches the description, while keeping accurate lip–audio sync and rhythm-aligned gestures. This mode is useful when a clean portrait is unavailable or when privacy constraints disallow using a real face; it does not aim to reproduce any specific person.

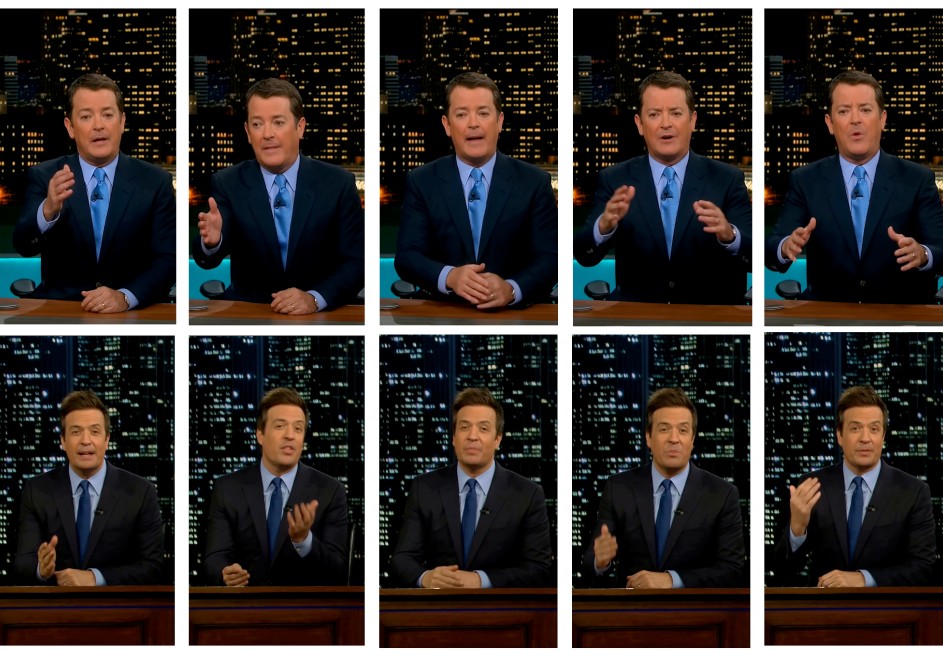

*A man in a dark suit, light blue shirt, and blue tie is speaking while making hand gestures. He is seated at a desk with a cityscape background visible behind him. The man moves his hands in various positions. The background remains consistent with tall buildings and illuminated windows, suggesting a nighttime city setting.*

Figure 8: **Reference-free generation from text + audio.** With only a prompt and audio, ApoAvatar creates a speaker consistent with the text and maintains clear lip sync and natural hand motions across the clip.

## A.3 MORE DEMO CASES

We provide additional qualitative results in Figure 9. The examples cover varied real-world settings, including indoor vlogs, outdoor interviews, desk-top recordings, and scenes with hand-held objects or microphones. ApoAvatar keeps identity and color stable, produces clear lips, and follows the audio rhythm with natural upper-body gestures, even under occlusions, rapid speaking styles, and long utterances. These cases show that our pose-free pipeline generalizes well across speakers, backgrounds, and camera setups.

## A.4 LLM STATEMENT

After completing the technical content of this paper, we used a large language model (LLM) as a writing assistant to proofread grammar, improve clarity, and harmonize style. The LLM did not generate research ideas, methods, experiments, analyses, or conclusions; all technical contributions, equations, figures, and results were authored by the authors. Every suggested edit was reviewed and approved by the authors, who take full responsibility for the final text. The LLM was not used to create or modify data, code, or citations; no references were introduced without verification. We shared only manuscript text (e.g., paragraphs and figure captions) with the LLM and did not upload proprietary datasets or identifying information. Our use of the LLM complies with the conference policy and institutional guidelines.

## A.5 LIMITATION

Our method targets single-person, audio-driven avatar animation. We do not yet handle multi-person scenes with turn-taking, overlapping speech, or person-specific audio–motion binding. Extending the framework to multiple speakers will require speaker-aware routing of audio features, per-person pose priors, and robust alignment across interacting subjects. We consider multi-person audio-driven animation an important direction for future work.

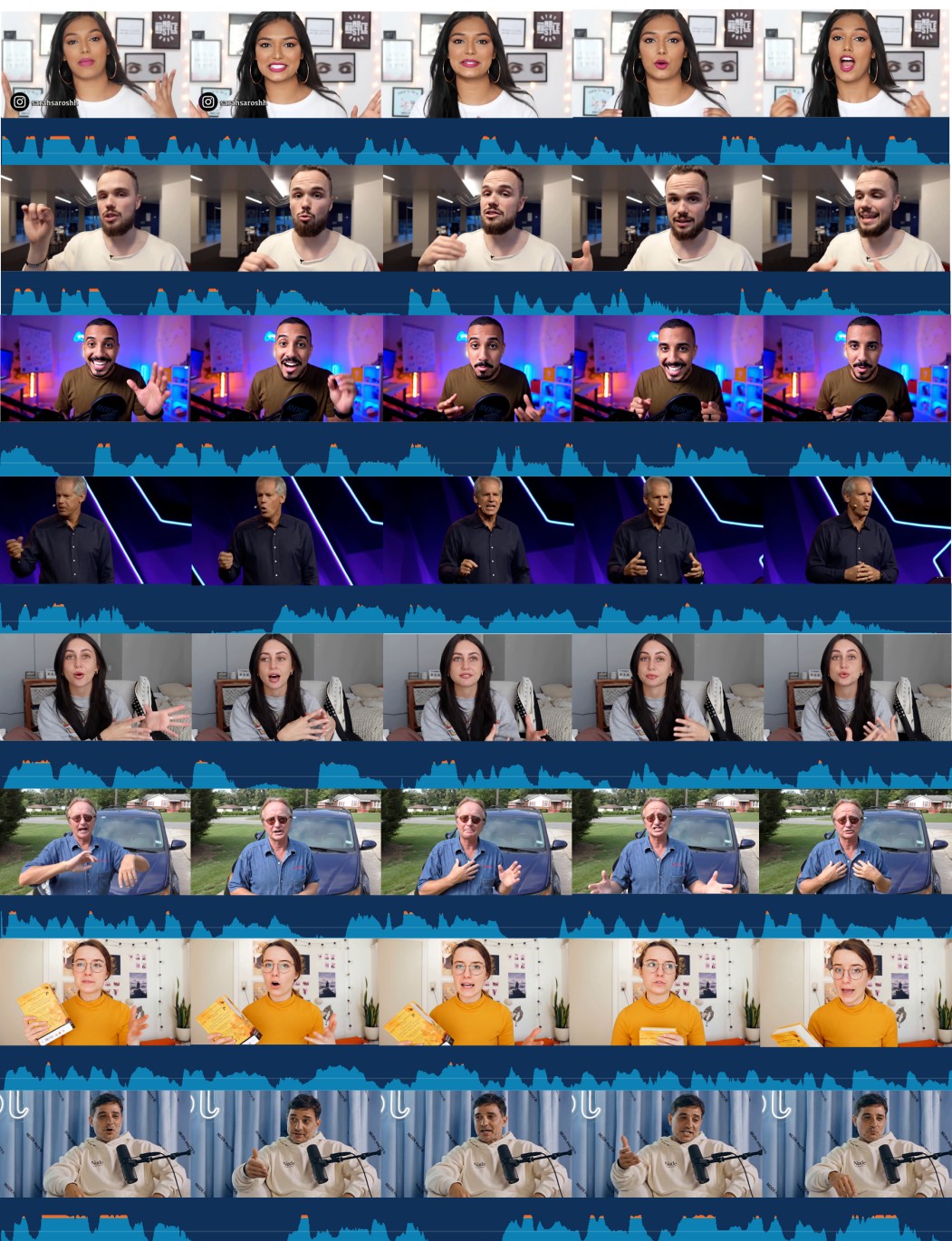

Figure 9: **More demo cases.** We present additional qualitative results across diverse scenarios, demonstrating the versatility of our ApoAvatar.

