# OpenReview forum: "ApoAvatar: Expressive Audio-Driven Avatar Generation via Refocused Audio-Pose Priors"
_ICLR.cc/2026/Conference — Submitted to ICLR 2026_

### Official Review · Reviewer_DSoK · 2025-10-20

**Soundness:** 2
**Presentation:** 3
**Contribution:** 3
**Rating:** 4
**Confidence:** 4

**Summary:**

This paper proposes ApoAvatar, a DiT-based framework for audio-driven video generation, aiming to address the common issue in existing methods where body motions are poorly synchronized with speech rhythm, resulting in stiff and unnatural animations. ApoAvatar introduces two key innovations: 1) an Audio-Pose Prior Refocusing mechanism that dynamically modulates the strength of pose guidance based on frame-level audio intensity—strong accents amplify gesture magnitude, while quiet segments suppress unnecessary motion, thus aligning gesture dynamics with speaking style. 2) a Frame-Wise Audio–Video Interaction module that employs a bidirectional cross-attention within an Audio DiT adapter, enabling audio features to be refined using the current visual context and the refocused pose prior, producing "pose-aware" audio embeddings. The framework supports unified inference with or without pose input. Experiments on the EMTD and HDTF datasets demonstrate that ApoAvatar outperforms baselines in lip-audio synchronization, gesture expressiveness, motion naturalness, and identity preservation.

**Strengths:**

* This paper identifies a critical yet often overlooked limitation in most existing works: audio and pose features are typically modeled as independently contributing modalities with insufficient interaction. To address this issue, the authors propose well-designed solutions, whose effectiveness is thoroughly validated through ablation studies. This work offers the community a novel and valuable perspective for optimizing audio-driven video generation.
* This work introduces an audio-aware pose prior refocusing mechanism that acts as a pose feature refiner. By dynamically modulating the retention level of pose embeddings based on prosodic cues, the method naturally scales gesture amplitude according to speech intonation. It amplifies movements during accented syllables and suppresses redundant motions during silent segments, thereby generating more expressive and rhythmically coherent full-body animations.
* Similarly, the paper designs a frame-wise audio-video interaction architecture that serves as an audio feature refiner. At each denoising step, the model updates audio representations by conditioning on both the current video state and the refocused pose prior, transforming audio features from static inputs into dynamically evolving, "video-aware" signals. This significantly enhances short-term synchronization and motion smoothness.

**Weaknesses:**

* The experimental setup is insufficiently described: Section 4.1 lacks essential details regarding the experimental configuration, including—but not limited to—the number of training stages, and whether full-parameter fine-tuning, LoRA, or SFT was used for parameter updates.
* Insufficient discussion on the text modality: The backbone model used in this work is an image-to-video (I2V) model, in which the text input inherently provides strong conditioning. However, the paper does not adequately discuss the role of the text branch, such as how the text labels in the training data are obtained or what specific text inputs are used during comparison with other methods. Notably, methods like MultiTalk[1] and OmniAvatar[2] do not use pose conditioning; hence, when text descriptions are insufficient, their generated body motions may naturally be less expressive. This oversight may unintentionally give the proposed method an unfair advantage in comparative evaluation.
* Qualitative comparison is inadequate: The paper only provides nine qualitative cases in the appendix, which do not even include the example shown in Figure 4. Moreover, while OmniAvatar[2] is included in quantitative comparisons, it is missing from the qualitative results. These issues raise concerns that the presented results may be cherry-picked, failing to fully demonstrate the effectiveness and superiority of the proposed approach.
* Concerns regarding reproducibility: The work builds upon Goku[3], a foundational model that is neither open-sourced nor commercially available. Despite initial promises by the Goku[3] authors to release it, the model remains inaccessible to the public as of now. This severely undermines the reproducibility of the work, as readers cannot verify whether the generated results stem from the base model's capabilities or from the contributions introduced in this paper. The experimental results would be significantly more convincing if the proposed method were implemented and evaluated on an open-source foundation model such as Wan[4], which is already used by several compared methods (e.g., FantasyTalking[5], MultiTalk[1], InfiniteTalk[6], OmniAvatar[2]).

[1]Kong, Zhe, et al. "Let Them Talk: Audio-Driven Multi-Person Conversational Video Generation." arXiv preprint arXiv:2505.22647 (2025).

[2]Gan, Qijun, et al. "OmniAvatar: Efficient Audio-Driven Avatar Video Generation with Adaptive Body Animation." arXiv preprint arXiv:2506.18866 (2025).

[3]Chen, Shoufa, et al. "Goku: Flow based video generative foundation models." Proceedings of the Computer Vision and Pattern Recognition Conference. 2025.

[4]Wan, Team, et al. "Wan: Open and advanced large-scale video generative models." arXiv preprint arXiv:2503.20314 (2025).

[5]Wang, Mengchao, et al. "Fantasytalking: Realistic talking portrait generation via coherent motion synthesis." arXiv preprint arXiv:2504.04842 (2025).

[6]Yang, Shaoshu, et al. "InfiniteTalk: Audio-driven Video Generation for Sparse-Frame Video Dubbing." arXiv preprint arXiv:2508.14033 (2025).

**Questions:**

* Regarding the training procedure: Could the authors clarify the overall training pipeline and detailed training configurations? Specifically, how are the text labels in the training data obtained, and how is the influence of text conditioning on pose generation balanced?
* Regarding comparisons: Are the text inputs kept consistent across all compared methods? Additionally, could the authors provide more generation results via an anonymous link to better support the claims in the paper?
* Role of pose conditioning: When pose guidance is available, how does the proposed method compare in performance against recent pose-driven approaches such as RealisDance-DiT[1] and X-UniMotion[2]?
* Evolution of the pose branch: Omni-Human1.5[3] deliberately removed pose conditioning in its upgrade from Omni-Human1[4], as challenges in body motion generation—such as body turning, finger articulation, and dance movements—remain difficult. In contrast, this work weakens the role of the text branch and reintroduces the pose branch. What deeper insights or design considerations motivate this architectural choice?

[1]Zhou, Jingkai, et al. "RealisDance-DiT: Simple yet Strong Baseline towards Controllable Character Animation in the Wild." arXiv preprint arXiv:2504.14977 (2025).

[2]Song, Guoxian, et al. "X-UniMotion: Animating Human Images with Expressive, Unified and Identity-Agnostic Motion Latents." arXiv preprint arXiv:2508.09383 (2025).

[3]Jiang, Jianwen, et al. "Omnihuman-1.5: Instilling an active mind in avatars via cognitive simulation." arXiv preprint arXiv:2508.19209 (2025).

[4]Lin, Gaojie, et al. "Omnihuman-1: Rethinking the scaling-up of one-stage conditioned human animation models." arXiv preprint arXiv:2502.01061 (2025).

---

> ### Author Response · Authors · 2025-11-22
> **Author Response to Reviewer DSoK (Part 1/3)**
>
> We are grateful for your positive review and valuable comments, and we hope our response fully resolves your concerns. For your convenience, we abbreviate *Weakness* and *Question* as W and Q in our rebuttal (e.g., W1 refers to the first weakness). Each response is clearly labeled to indicate which W or Q it addresses.
>
> ***
>
> 1. **Experimental Setup and Training Procedure (W1 & Q1)**
>
>
> Thank you for pointing out the missing training details. We have updated Section 4.1 with a clearer description of the full training pipeline.
>
> **Two-stage training.**  Our method is trained in two stages, each with different learning objectives and learning rates:
>
> - Stage 1 (lr = 1e-4). We jointly train the four branches (audio, pose, text, and image/video). This forces the model to learn an effective division of labor:
> 	- The text and video branches focus on high-level semantic structure,
>
> 	- The audio and pose branch naturally specializes in lip sync and speech-driven motion patterns. This stage establishes strong modality-specific priors.
>
> - Stage 2 (lr = 3e-5). We construct the final model by initializing the text and video branches with their original pre-trained weights and initializing the audio branch with the warmed-up weights from Stage 1. This avoids modality conflict and ensures that each branch preserves its intended conditioning power, resulting in a balanced interaction between text conditioning and pose generation.
>
>
> **Text labels.**
> For the training data, we generate text descriptions using **Tarsier2** \[1\], followed by a simple prompt describing the speaking scenario, such as *“The person is speaking toward the camera, with body movements naturally synchronized to their speech.”* This process is now described in the revised manuscript.
>
> **Training and inference configuration.**  We train with a global batch size of 256, and generate clips at 24 fps with 720p resolution (short side) for all experiments. The model is trained with full-parameter fine-tuning.
> We hope these clarifications resolve the reviewer’s concerns, and we have incorporated all details into the updated Section 4.1.
>
> ***
>
> 2. **Text Modality and Fairness in Comparison (W2 & Q2)**
>
>
> Thank you for highlighting this important point. We apologize for not explaining the text modality clearly in the initial submission. In all experiments, **the same text prompt is used for every compared method**, ensuring full fairness and preventing any advantage from stronger or more detailed textual conditioning.
>
> **Text label generation.**  For the training data, we generate text descriptions using **Tarsier2** \[1\], followed by a simple prompt describing the speaking scenario. For example:
>
> > *“The image shows a person wearing a dark suit, a light blue dress shirt, and a blue tie, sitting at a desk. The background features a cityscape with illuminated windows, suggesting a nighttime setting. He is gesturing with* his *hands while talking. The person is speaking toward the camera, with body movements naturally synchronized to their speech.”*
>
> This captioning pipeline is now documented in the revised manuscript.
>
> **Fair comparison.**  During evaluation, **all competing methods receive exactly the same text input**, including MultiTalk and OmniAvatar. Therefore, differences in expressiveness cannot be attributed to text conditioning, and no method receives any advantage or extra supervision.

---

> ### Author Response · Authors · 2025-11-22
> **Author Response to Reviewer DSoK (Part 2/3)**
>
> 3. **More Qualitative Comparison (W3)**
>
>
> Thank you for raising this important point. We acknowledge that the qualitative results included in the appendix were limited. This is primarily due to the strict 100MB size limit imposed by ICLR for supplementary materials, which prevented us from including a larger set of examples in the initial submission.
>
> To address this concern, we now provide additional qualitative results, including the example shown in Figure 4 and samples for OmniAvatar. These supplementary videos are available at the link: [More Comparison 1](https://anonymousdemo2026.github.io/anonymous-iclr26-demo/static/More\_Comparison1.mp4), [More Comparison 1](https://anonymousdemo2026.github.io/anonymous-iclr26-demo/static/More\_Comparison2.mp4), [Fig4 Demo](https://anonymousdemo2026.github.io/anonymous-iclr26-demo/static/Fig4\_demo.mp4).
>
> In addition, to further strengthen the evaluation, we conducted complete quantitative comparisons for the OmniHuman series. Since OmniHuman 1.0 and 1.5 are closed-source and only available through an online interface, running all test samples requires executing each case individually. Despite the substantial manual effort involved, we have obtained the full evaluation and incorporated the results into the updated tables.
>
> | **Method** | **FID↓** | **FVD↓** | **IQA↑** | **ASE↑** | **SYNC↑** | **SYND↓** | **FSIM↑** | **HKC↑** | **HKV** |
> | :-- | :-- | :-- | :-- | :-- | :-- | :-- | :-- | :-- | :-- |
> | Fantasy | 133.73 | 1307.20 | 2.11 | 1.12 | 1.11 | 12.88 | 0.59 | 0.57 | 8.0 |
> | HyaV | 139.39 | 2160.92 | 1.76 | 1.18 | 4.89 | 9.37 | 0.67 | 0.75 | 29.2 |
> | Hallo3 | 104.51 | 1256.10 | 2.31 | 1.48 | 4.26 | 10.22 | 0.73 | 0.77 | 6.3 |
> | MultiTalk | 103.68 | 1040.43 | 2.07 | 1.30 | 6.34 | 8.47 | 0.71 | 0.79 | 14.6 |
> | OmniAvatar | 82.54 | 1104.99 | 2.16 | 1.31 | 5.4 | 9.13 | 0.72 | 0.86 | 28.7 |
> | Omni-human 1.0 | 99.17 | 1090.28 | 2.22 | 1.42 | 6.54 | 8.55 | 0.81 | 0.84 | 8.7 |
> | Omni-human 1.5 | 108.54 | 1132.09 | 2.17 | 1.44 | 6.32 | 8.81 | 0.75 | 0.81 | 12.9 |
> | Ours | **67.11** | **1020.54** | **2.31** | **1.65** | **6.88** | **8.17** | **0.86** | **0.93** | 14.5 |
>
> Across both the newly added qualitative examples and the expanded quantitative evaluations, our method consistently outperforms existing approaches, demonstrating advantages in motion naturalness, identity consistency, and audio alignment.
> We hope these additional results address your concerns and provide a more complete picture of the method's effectiveness.
>
> ***
>
> 4. **Comparison with RealisDance-DiT and X-UniMotion (Q3)**
>
>
> Thank you for the insightful question. Our method is fundamentally designed as an audio-driven generator for single-person, camera-facing speaking scenarios. In other words, our pose conditioning supports simple camera-facing pose control, but does not aim to handle complex full-body or large-range articulated motions.
> In contrast, RealisDance-DiT and X-UniMotion target a very different problem setting: expressive, full-body human animation with large pose variations, turning motions, limb articulation, and dancing movements. Our model is not designed or trained for such scenarios, mainly because:
>
> (1) **Dataset mismatch.**  To maintain high-quality lip-sync and speech alignment, our training data focuses on talking scenarios and does not contain sufficient complex action sequences.
>
> (2) **Different research objectives**. These two methods aim at general, expressive human animation, whereas we focus on stable and natural motion for single-person talking videos, which require a different modeling emphasis.
>
> We appreciate the reviewer for highlighting these influential works. Both RealisDance-DiT and X-UniMotion have been cited in the Introduction as important pose-driven baselines, and their insights are valuable for our planned future extension toward more complex motion control.

---

> ### Author Response · Authors · 2025-11-22
> **Author Response to Reviewer DSoK (Part 3/3)**
>
> 5. **Concerns related to Goku (W4)**
>
>
> Thank you for raising this concern. While the Goku model is not open-sourced, its architecture and training design are fully public, and we use the Goku-8B variant, whose VBench performance (84.85) is comparable to Wan-2.1. This places our backbone on the same capability tier as widely used open-source models, meaning that comparisons with Wan-based approaches are not advantaged by an unusually strong foundation model.
>
> Importantly, the effectiveness of our contributions is independent of the backbone. Our ablation studies (Tables 4 and 5) explicitly isolate the effects of the refocusing and interaction modules, and a reviewer eXxq has already noted this point as “Strong ablation support: Tables 4–5 isolate contributions of refocusing and interaction modules.” This external confirmation further strengthens that the improvements stem from our method rather than from Goku itself.
>
> A full Wan-based re-implementation is not possible within the rebuttal timeframe, as it would require retraining and re-running extensive quantitative and qualitative evaluations. However, we are willing to explore an open-source version based on Wan as part of our ongoing work. In addition, we plan to provide public online access to our model for the community. These steps will further enhance accessibility going forward.
>
> ***
>
> 6. **Discussion about the Pose Branch (Q4)**
>
>
> Thank you for this thoughtful question. Our design has a different motivation from OmniHuman-1.5. OmniHuman-1.5 removes pose conditioning to focus on the MLLM-based prompt-rewriting module and stronger text alignment. However, this often leads to identity inconsistency, which we can clearly observe in our comparison demos above.
>
> In our case, the target scenario is single-person, camera-facing talking videos, where subtle upper-body and facial cues matter a lot. For this setting, pose information is highly valuable. The Sapiens pose extractor provides dense and reliable keypoints (see our pose-visualization [Demo](https://anonymousdemo2026.github.io/anonymous-iclr26-demo/static/pose\_label.mp4)), including mouth, arms, fingers, pupils, and cheeks. These cues help generate more natural gestures and also keep the facial identity more stable.
>
> With such accurate pose labels, we can learn a strong audio–pose prior that improves both realism and identity consistency. This effect is consistently supported by (1) qualitative comparisons, (2) user-study results, and (3) quantitative evaluations, all showing that models trained with pose information (e.g., OmniHuman-1.0 and ours) achieve higher identity consistency than OmniHuman-1.5.
>
> Therefore, reintroducing the pose branch fits our target domain of full-body communication and strengthens naturalness, rather than prioritizing text-level control as in OmniHuman-1.5. This design motivation has also been positively recognized by Reviewer 7rdx.
>
> ***
>
> \[1\] Yuan, Liping, et al. "Tarsier2: Advancing large vision-language models from detailed video description to comprehensive video understanding." *arXiv preprint arXiv:2501.07888* (2025).

---

> > ### Comment · Reviewer_DSoK · 2025-11-27
> > **Official Comment by Reviewer DSoK**
> >
> > Thank you for the detailed clarifications and the expanded experiments. The authors’ responses have addressed some of my concerns, and I am raising my score to 6. Finally, I strongly encourage the authors to open-source their code or provide accessible channels (e.g., a demo) for readers to experience the actual performance of this work. In the era of large models, closed-source papers with only static results contribute limited value to the community—readers often cannot fully reproduce the work or truly appreciate the impact of the proposed innovations.

---

> > > ### Author Response · Authors · 2025-11-27
> > >
> > > We sincerely appreciate your positive feedback and the higher score. We fully share your view on the importance of open research. We are committed to providing interactive access, more demos, and detailed code to support reproducibility and enable the community to experience our work firsthand.

---

### Official Review · Reviewer_7rdx · 2025-10-23

**Soundness:** 4
**Presentation:** 4
**Contribution:** 3
**Rating:** 8
**Confidence:** 5

**Summary:**

To address the issue that avatar poses in current talking head generation tasks are overly rigid and lack vividness, this paper proposes an "Audio-Pose Prior Refocusing Mechanism" and a "Frame-Wise Audio–Video Interaction Strategy" to enhance the modeling between audio and motion, thereby achieving more vivid avatar generation results. The paper demonstrates the effectiveness of the proposed scheme through extensive quantitative and qualitative experiments as well as user studies.

**Strengths:**

- The paper has a clear research motivation and well-organized presentation. Additionally, it conducts extensive experiments and compares with numerous recent baselines, which is highly convincing, demonstrating sufficient and solid research efforts.
- Rigid movements in digital human driving are a practical problem, and the paper proposes an effective solution to tackle this issue.
- The paper presents a highly general talking head generation framework that supports full-body driving. Compared with recent methods focusing only on the head, this framework holds greater practical significance.

**Weaknesses:**

- The paper points out the lack of interaction between pose and audio, stating that "audio, pose, and visual context are often merged by simple concatenation or a single round of attention." This issue seems to target single-stream DiT. Does this problem still exist for dual-stream (or multi-stream) MM-DiT? Since dual-stream/multi-stream designs inherently introduce interactions between modalities.
- Using accents as the prior for pose is reasonable; however, this approach of leveraging inductive bias may make it difficult to learn certain modes that do not rely on this prior.

**Questions:**

Please refer to the weaknesses.

---

> ### Author Response · Authors · 2025-11-22
>
> We sincerely thank the reviewer for the highly constructive and insightful comments, which greatly helped us strengthen both our analysis and technical clarity.
>
> 1. **Interactions in MM-DiT architectures**
>
>
> Thank you for the insightful comment. Introducing audio as an additional stream in a multi-stream MM-DiT is indeed a possible direction. However, adding an audio stream greatly increases the number of transformer blocks, parameters, and inter-stream attention operations. This makes training more expensive and less stable. It also requires much stronger multi-GPU parallelism and infrastructure support. The benefit is not guaranteed because audio tokens are short and sparse compared with visual tokens. A heavy multi-stream design may still fail to capture fine audio–motion timing cues.
>
> Regarding the interaction issue: although dual- or multi-stream MM-DiT models provide richer interactions, current video backbones mainly focus on interactions between text tokens and visual tokens. In many MM-DiT-based methods, audio and pose are still injected in simple ways, such as token concatenation, additive conditioning, or cross-attention.
>
> Our Audio–Pose Prior Refocusing mechanism is designed to address this limitation. Instead of modifying or expanding the backbone, we add a lightweight frame-level alignment module that strengthens the correlation between audio and pose during training. This improves audio–motion interaction while keeping the backbone unchanged. It is efficient, stable, and compatible with existing architectures.
>
> ***
>
> 2. **About the inductive bias of using audio accents**
>
>
> We appreciate the reviewer’s thoughtful comment. Using accents as an inductive prior can indeed be risky if not handled carefully. To reduce this risk, we use two simple strategies:
>
> **Pose dropout.**  We use two forms of dropout. First, with a probability of 0.5, the entire pose input is set to null, preventing the model from relying only on accent-driven cues. Second, our refocusing mechanism adds a frame-level dropout that depends on the audio intensity: pose cues are kept around stressed speech segments and dropped more often during flat or silent segments. These two forms of dropout encourage the model to learn audio-aligned motion patterns.
>
> **Two-stage training.**
> To avoid overfitting to this inductive bias, we adopt a two-stage training strategy. In **Stage 1** (lr = 1e-4), we jointly train the four branches (audio, pose, text, and image/video), so that text+video focus on high-level semantics while the audio/pose branches specialize in lip sync and motion patterns. In **Stage 2** (lr = 3e-5), we re-initialize the text and video branches with their original pre-trained weights and keep the warmed-up audio/pose weights from Stage 1, then fine-tune the whole model. This preserves strong, diverse motion priors and prevents the accent-based prior from suppressing motion modes that do not strictly follow audio intensity.
>
> As shown in the walking-sequence demos:
>
> - [Demo A](https://anonymousdemo2026.github.io/anonymous-iclr26-demo/static/walking.mp4) – Text-controlled lateral walking while speaking
>
> - [Demo B](https://anonymousdemo2026.github.io/anonymous-iclr26-demo/static/moving.mp4) – Standing-up transition from a seated pose while maintaining speech coherence
>
> The model still generates **natural and diverse body poses**, suggesting that the accent-based prior does not reduce motion variety in practice.

---

> > ### Comment · Reviewer_7rdx · 2025-11-24
> >
> > Thanks for the authors' reply. My concern has been well addressed, and I will keep the original score.

---

> > > ### Author Response · Authors · 2025-11-26
> > >
> > > Thank you once again for your valuable suggestions and positive feedback. We are glad that our response has addressed your concerns. We truly appreciate your support and your insightful comments.

---

### Official Review · Reviewer_eXxq · 2025-10-30

**Soundness:** 3
**Presentation:** 3
**Contribution:** 2
**Rating:** 6
**Confidence:** 2

**Summary:**

This paper addresses the challenge of generating expressive audio-driven human avatars by improving audio-motion coupling. The authors propose ApoAvatar, a diffusion-based framework featuring an Audio-Pose Prior Refocusing mechanism that adjusts pose embeddings based on frame-level audio intensity. A Frame-Wise Audio-Video Interaction module refines audio features using visual context and refocused pose priors via bidirectional cross-attention. The model supports both pose-conditioned and pose-free inference. Evaluations on EMTD and HDTF datasets demonstrate improvements in lip synchronization, gesture expressiveness, and motion naturalness.

**Strengths:**

1. Novel audio-prosody modulation: The Audio-Pose Prior Refocusing dynamically scales gesture intensity using RMS-based loudness, addressing audio-motion decoupling.

2. Effective multimodal interaction: Frame-Wise Cross-Attention enables audio refinement via video/pose context, improving short-term synchronization.

3. Flexible inference design: Decoupled classifier-free guidance allows per-modality control, and pose-free inference maintains robustness.

4. Rigorous evaluation: Comprehensive metrics and user studies validate superiority in lip sync and motion naturalness.

5. Strong ablation support: Tables 4-5 isolate contributions of refocusing and interaction modules.

**Weaknesses:**

1. Superficial hand motion analysis: Hand stability is evaluated only via HKC/HKV, lacking perceptual metrics or qualitative examples of failure cases.

2. Incomplete baseline comparison: OmniHuman’s exclusion from main tables limits quantitative context, despite its relevance as a state-of-the-art method.

3. Underexplored failure modes: Occlusion robustness is mentioned in Appendix Fig. 7 but not quantitatively analyzed or discussed in the main text.

4. Limited dataset diversity: Training uses curated single-person clips; generalization to complex scenes, such as multi-person interactions, is unverified.

5. Ambiguous audio alignment details: The hop length h and sampling rate sr for RMS calculation are unspecified, hindering reproducibility.

**Questions:**

1. Can you include OmniHuman’s quantitative results (e.g., FID/SYNC) on a subset of EMTD/HDTF compatible with its 15-second input limit? Because OmniHuman is a key SOTA comparator (Sec 4.2). Partial results would contextualize claims of superiority more fairly.

2. For pose-free inference under occlusion in Appendix Fig. 7, report metrics of FID and FSIM on occluded vs. clean samples. How does refocusing mitigate pose errors? The claim of robustness lacks quantitative support.

3. You state pose is "heavily dropped during training" in Sec 3.4. What was the dropout rate? If pose is dropped >40% of frames, how does this impact gesture diversity??Because High dropout may explain limited motion variety in baselines as shown in Fig. 5.

4. The author mentions that the amplitude of the motion is coupled with the intensity of the audio. However, the types of motions are essentially limited to a single vertical hand gesture. In comparison, OmniHuman-1.5 possesses a greater variety of hand gestures, offering more diversity. It remains unclear whether the gesture motions can be generalised to encompass a wider range.

5. The author claims mentions that the model can ensure head-stable positioning, but is this truly an aspect requiring improvement? Appropriate head rotation should render video subjects appear more natural. Might the article be solely focused on hand poses while overlooking the coupling between the head, gaze, and audio content?

---

> ### Author Response · Authors · 2025-11-22
> **Author Response to Reviewer eXxq (Part 1/3)**
>
> We are grateful for your positive review and valuable comments, and we hope our response fully resolves your concerns. For your convenience, we abbreviate *Weakness* and *Question* as W and Q in our rebuttal (e.g., W1 refers to the first weakness). Each response is clearly labeled to indicate which W or Q it addresses.
>
> ***
>
> 1. **Perceptual metrics (W1)**
>
>
> **Evaluation of audio–motion alignment**.
> Thank you for the insightful comment. Our user study already includes an explicit *Audio–Pose Alignment* criterion, where participants consistently rate our method higher than all baselines (Table 3). This provides direct human-perceived evidence of better audio-driven motion consistency.
> To further support this result, we report an F1-based alignment analysis. We extract audio events by combining onset detection and RMS energy peaks (0.6×onset + 0.4×energy). Motion events are obtained by tracking hand keypoints (MediaPipe) and locating local maxima in motion amplitude. An audio and motion event is considered matched if they fall within a 200-ms window.
> Precision is defined as the proportion of motion events that have a corresponding audio event, while recall measures the proportion of audio events that are matched by a motion event; their harmonic mean gives the final F1 score. Evaluated on the EMTD test set:
>
> | Methods | Multitalk | OmniAvatar | Omni-human1.0 | Omni-human1.5 | Ours |
> | --- | --- | --- | --- | --- | --- |
> | F1-score | 0.478 | 0.445 | 0.551 | 0.489 | 0.828 |
>
> Our model achieves an F1 score of 0.828, demonstrating strong temporal consistency between audio semantics and generated hand motion. This analysis complements the user-study results.
>
> ***
>
> 2. **Missing OmniHuman baseline (W2 & Q1)**
>
>
> We have added OmniHuman’s quantitative results on the EMTD test set to the revised version in Table 2 to ensure a fair and complete comparison. Since OmniHuman 1.0 and 1.5 are closed-source and only accessible through an online interface, obtaining the full set of quantitative results required running each test case individually. Despite the substantial manual effort involved, we completed the full evaluation and incorporated the results into the tables below.
>
> | **Method** | **FID↓** | **FVD↓** | **IQA↑** | **ASE↑** | **SYNC↑** | **SYND↓** | **FSIM↑** | **HKC↑** | **HKV** |
> | :-- | :-- | :-- | :-- | :-- | :-- | :-- | :-- | :-- | :-- |
> | Fantasy | 133.73 | 1307.20 | 2.11 | 1.12 | 1.11 | 12.88 | 0.59 | 0.57 | 8.0 |
> | HyaV | 139.39 | 2160.92 | 1.76 | 1.18 | 4.89 | 9.37 | 0.67 | 0.75 | 29.2 |
> | Hallo3 | 104.51 | 1256.10 | 2.31 | 1.48 | 4.26 | 10.22 | 0.73 | 0.77 | 6.3 |
> | MultiTalk | 103.68 | 1040.43 | 2.07 | 1.30 | 6.34 | 8.47 | 0.71 | 0.79 | 14.6 |
> | OmniAvatar | 82.54 | 1104.99 | 2.16 | 1.31 | 5.4 | 9.13 | 0.72 | 0.86 | 28.7 |
> | Omni-human 1.0 | 99.17 | 1090.28 | 2.22 | 1.42 | 6.54 | 8.55 | 0.81 | 0.84 | 8.7 |
> | Omni-human 1.5 | 108.54 | 1132.09 | 2.17 | 1.44 | 6.32 | 8.81 | 0.75 | 0.81 | 12.9 |
> | Ours | **67.11** | **1020.54** | **2.31** | **1.65** | **6.88** | **8.17** | **0.86** | **0.93** | 14.5 |
>
> The results consistently show that our method achieves the best overall performance across video quality, lip sync, identity preservation, and hand stability, further validating the effectiveness of our design.
>
> ***
>
> 3. **Metrics of FID and FSIM on occluded vs. clean samples (W3 & Q2)**
>
>
> Thank you for raising this question, and we apologize for the confusion caused by Appendix Fig. 7. The intention of this figure is to illustrate that our default evaluation uses pose-free inference because, when explicit pose inputs are provided at test time, occlusions may degrade the quality of the extracted keypoints. As a result, pose-conditioned inference may introduce instability, including facial distortion under heavy occlusion. This is precisely why, during training, our method relies on frame-wise audio–pose refocusing with dynamic pose dropout to build a soft audio–pose prior rather than a strict pose-control mechanism.
> To quantitatively evaluate robustness under occlusion, we collected **50 samples** with face occlusion and measured FID and FSIM:
>
> | Condition | FID ↓ | FSIM ↑ |
> | :-: | :-: | :-: |
> | Occluded | 75.19 | 0.91 |
> | Clean | 67.11 | 0.93 |
>
> As shown above, **the performance degradation under occlusion is small**, confirming that the model remains stable without catastrophic failure. Because the audio–pose prior acts as a *soft* guidance signal instead of a hard constraint, the model naturally falls back to **audio-driven motion** when pose cues are unreliable. This mitigates pose errors and explains why pose-free inference is more robust in occluded scenarios.

---

> ### Author Response · Authors · 2025-11-22
> **Author Response to Reviewer eXxq (Part 2/3)**
>
> 4. **Limited dataset diversity: generalization to multi-person scenes (W4)**
>
>
> Thank you for pointing out this limitation. We would like to clarify that multi-person audio-driven human video generation is a fundamentally different research setting, not merely a matter of collecting more data. Beyond video diversity, it requires addressing how multiple independent audio streams should be correctly associated with different spatial regions and identities within the video. This involves non-trivial architectural changes and is outside the scope of our single-person audio-driven framework.
>
>
> As stated in our Appendix, handling multi-person scenarios is indeed listed as part of our future work.
> However, following your suggestion, we conducted an initial exploration. Inspired by HunyuanVideo-Avatar \[1\], we manually introduced per-person face masks during generation and routed each audio track to the corresponding facial region. Using this approach, we obtained preliminary dual-person dialogue results, as shown in the demo: [Multi-Person Demo](https://anonymousdemo2026.github.io/anonymous-iclr26-demo/static/Multiperson\_trial.mp4). These results demonstrate that our model can produce **reasonable multi-person dialogue animations**, although the approach relies on external face masks and pre-assigned audio–region mappings. While not a complete multi-person solution, it serves as a practical proof-of-concept and confirms that extending the method to multi-person settings is feasible.
>
> ***
>
> 5. **Audio alignment details (W5)**
>
>
> Thank you for pointing this out. For RMS-based audio analysis, we use hop length h=320 and a sampling rate sr=16kHz. After computing the RMS sequence, we apply temporal interpolation to match the video frame rate and then ensure frame-wise alignment with the corresponding pose frames.
>
> ***
>
> 6. **Pose dropout rate and its influence on gesture diversity (Q3)**
>
>
> Thank you for the question. Our training uses two complementary dropout mechanisms:
>
> **Whole-pose dropout (fixed)**: With a probability of 0.5, the entire pose sequence is set to null.
>
> **Frame-level dropout (dynamic)**: This is part of our Audio-Pose Refocusing mechanism. The dropout probability varies with the audio rhythm and emphasis:
>
> - Around accented or stressed phonemes, pose inputs are retained and serve as effective control signals.
>
> - During flat speech segments or silence, pose inputs are dropped more frequently.
>
>
> This design follows natural human communication patterns, people exhibit stronger gestures near emphasized speech and weaker gestures during neutral segments. By training under this dynamic dropout, the model learns an implicit audio–pose prior, enabling it to generate stable and natural motion even without pose inputs at inference. Importantly, this mechanism does not reduce gesture diversity. Instead, it helps the model learn when gestures should or should not occur, rather than copying pose motions blindly.
> In fact, the examples in Fig. 5 and the accompanying demo [Refocusing Ablation](https://anonymousdemo2026.github.io/anonymous-iclr26-demo/static/refocusing\_ablation.mp4) show clear variations in hand motions, body movements, demonstrating that the model can still generate diverse, contextually appropriate gestures.
>
> 7. **Gesture diversity (Q4)**
>
>
> Thank you for the insightful observation. Unlike OmniHuman-1.5, which relies on *agent-based control* to explicitly inject a large variety of gestures (sometimes at the cost of identity shift), our design emphasizes stability and naturalness.
> Despite the reviewer’s concern, our model is not limited to a single vertical hand gesture. As shown in the demos below:
>
> - [Demo A](https://anonymousdemo2026.github.io/anonymous-iclr26-demo/static/walking.mp4) – Text-controlled lateral walking while speaking
>
> - [Demo B](https://anonymousdemo2026.github.io/anonymous-iclr26-demo/static/moving.mp4) – Standing-up transition from a seated pose while maintaining speech coherence
>
> Our system generates rich and contextually appropriate upper-body motions even without pose conditioning at inference. These results demonstrate that the learned audio–pose prior generalizes beyond vertical gestures and can support a broader range of natural human motions.

---

> ### Author Response · Authors · 2025-11-22
> **Author Response to Reviewer eXxq (Part 3/3)**
>
> 8. **Concerns about natural head movement (Q5)**
>
>
> We appreciate the question. Our model does not aim to suppress head motion. The pose representation extracted includes mouth, jawline, arms, hands, fingers, pupils, and cheek movements, providing a fine-grained full-head and upper-body signal (visualized in the [Demo](https://anonymousdemo2026.github.io/anonymous-iclr26-demo/static/pose\_label.mp4)).
> The purpose of emphasizing “head-stable positioning” is to prevent unnatural global drift, which is a common failure mode in audio-driven generation when explicit pose is absent or noisy. Importantly, the model still produces natural head rotations and gaze dynamics driven by prosody and expression in the input audio.
>
> ***
>
> \[1\] Chen, Yi, et al. "HunyuanVideo-Avatar: High-Fidelity Audio-Driven Human Animation for Multiple Characters." *arXiv preprint arXiv:2505.20156* (2025).

---

> ### Author Response · Authors · 2025-11-27
>
> Dear Reviewer eXxq,
>
> As the discussion period deadline approaches, we would like to kindly ask whether you have any thoughts on the rebuttal of our manuscript and whether it has influenced your assessment. We would be happy to address any remaining questions or concerns.
>
> Best regards,
> The Authors

---

### Official Review · Reviewer_itA1 · 2025-11-01

**Soundness:** 2
**Presentation:** 3
**Contribution:** 2
**Rating:** 4
**Confidence:** 4

**Summary:**

This paper presents a novel framework for audio-driven human video generation. Specifically, it introduces an Audio-Pose Prior Refocusing mechanism to enhance the coupling between audio and motion, and a Frame-Wise Audio–Video Interaction module that refines audio features by leveraging both the visual context and the refocused pose prior. Extensive experiments demonstrate the effectiveness of the proposed approach, showing promising results.

**Strengths:**

1. The Audio-Pose Prior Refocusing mechanism is novel and interesting.

2. The paper’s presentation is clear and well-structured.

3. Comparisons with prior work are fair, thorough, and self-contained.

**Weaknesses:**

1. The Frame-Wise Audio–Video Interaction may accumulate errors if the refocused pose prior is inaccurate. Additionally, no visual ablation studies are provided to support the design choices.

2. The identity used in the first figure is not appropriate, as it appears overly sexualized; more formal examples should be used.

3. The supplementary material is incomplete, lacking videos corresponding to all figures in the main text and appendix, which is insufficient.

4. The current evaluation metrics do not adequately assess body motion and audio alignment, despite this being a major claim of the paper.

**Questions:**

How is the pose acquired when only audio is provided?

---

> ### Author Response · Authors · 2025-11-22
>
> We are grateful for your positive review and valuable comments, and we hope our response fully resolves your concerns.
>
> ***
>
> 1. Pose Accuracy & Visual Ablations
>
> **Regarding Pose Reliability:**  To minimize error propagation, we employ the state-of-the-art Sapiens model \[1\] (ECCV 2024 Best Paper Candidate) for pose extraction. As demonstrated in our supplementary video [Pose Label Demo](https://anonymousdemo2026.github.io/anonymous-iclr26-demo/static/pose\_label.mp4)
> , the extracted poses are highly accurate and stable, providing a robust foundation for the Audio–Pose prior.
>
> **Visual Ablation Study:**  Following your suggestion, we have provided a visual comparison at [Frame-wise Ablation](https://anonymousdemo2026.github.io/anonymous-iclr26-demo/static/frame-wise\_ablation.mp4).
> The results clearly indicate that the Frame-Wise Audio-Video Interaction module is crucial for generating expressive motion, while the model without it tends to produce flatter results.
>
> ***
>
> 2. **Inappropriate identity**
>
>
> We sincerely appreciate this feedback and have prepared an alternative, formal example to replace the identity in Figure 1. Please refer to the new demo below: [Formal Case](https://anonymousdemo2026.github.io/anonymous-iclr26-demo/static/formal\_demo.mp4).
> If you find this suitable, we will ensure it is incorporated into the final version of the manuscript.
>
> ***
>
> 3. **Incomplete supplementary videos**
>
>
> Due to the **100MB size limit** for ICLR supplementary material, we could only include a subset of visual results. An extended set of demos is available at the following link: [video1](https://anonymousdemo2026.github.io/anonymous-iclr26-demo/static/no\_ref\_image.mp4), [video2](https://anonymousdemo2026.github.io/anonymous-iclr26-demo/static/whole\_sup\_demo.mp4), [video3](https://anonymousdemo2026.github.io/anonymous-iclr26-demo/static/pose\_free\_res.mp4).
>
> ***
>
> 4. **Evaluation of audio–motion alignment**
>
>
> Thank you for the insightful comment. Our user study already includes an explicit *Audio–Pose Alignment* criterion, where participants consistently rate our method higher than all baselines (Table 3). This provides direct human-perceived evidence of better audio-driven motion consistency.
> To further support this result, we report an F1-based alignment analysis. We extract audio events by combining onset detection and RMS energy peaks (0.6×onset + 0.4×energy). Motion events are obtained by tracking hand keypoints (MediaPipe) and locating local maxima in motion amplitude. An audio and motion event are considered matched if they fall within a 200-ms window.
> Precision is defined as the proportion of motion events that have a corresponding audio event, while recall measures the proportion of audio events that are matched by a motion event; their harmonic mean gives the final F1 score. Evaluated on the EMTD test set:
>
> | Methods | Multitalk | OmniAvatar | Omni-human1.0 | Omni-human1.5 | Ours |
> | --- | --- | --- | --- | --- | --- |
> | F1-score | 0.478 | 0.445 | 0.551 | 0.489 | 0.828 |
>
> Our model achieves an F1 score of 0.828, demonstrating strong temporal consistency between audio semantics and generated hand motion. This analysis complements the user-study results.
>
> > How is the pose acquired when only audio is provided?
>
> Our method does not require an explicit pose input when only audio is provided. The model has been trained to infer natural body movements directly from audio, using the learned audio–motion priors. Therefore, at inference time, supplying audio alone is sufficient for generating plausible pose sequences.

---

> ### Author Response · Authors · 2025-11-27
>
> Dear Reviewer itA1,
>
> As the discussion period deadline approaches, we would like to kindly ask whether you have any thoughts on the rebuttal of our manuscript and whether it has influenced your assessment. We would be happy to address any remaining questions or concerns.
>
> Best regards,
> The Authors

---

### Official Review · Reviewer_khjF · 2025-11-01

**Soundness:** 3
**Presentation:** 3
**Contribution:** 3
**Rating:** 6
**Confidence:** 5

**Summary:**

In this paper, the authors proposed a Audio-Pose Prior Refocusing mechanism that explicitly models prosody as a frame-level control signal to adaptively modulate pose embeddings and a Frame-Wise Audio–Video Interaction strategy in which an Audio DiT adapter refines audio using the current video context and the refocused pose prior.

**Strengths:**

The Frame-Wise Audio–Video Interaction strategy is interesting, where an Audio DiT adapter refines audio using the current video context and the refocused pose prior, producing pose and video-aware audio features that strengthen audio–motion coupling.

**Weaknesses:**

Major
- The quantitative numbers are not too encouraging, the improvement is marginal in many metrics. At multiple places bold numbers are wrongly marked e.g. Table 4 and 5 FVD values.
- The visual comparison results in the supplementary are limited, under more demo cases folder what are the inputs?, is not clear.
- Almost in all cases the person is in front of static background. What about those scenarios where e.g. person is giving a talk walking left right on the stage.

Minor
- Consider citing relevant works like DisFlowEm : One-Shot Emotional Talking Head Generation using Disentangled Pose and Expression Flow-Guidance WACV 2025

**Questions:**

Please refer to the weakness section

---

> ### Author Response · Authors · 2025-11-22
>
> We are grateful for your positive review and valuable comments, and we hope our response fully resolves your concerns.
>
> 1. **Quantitative results and FVD values**
>
>
> Our quantitative experiments are conducted on two datasets: HDTF and EMTD. Our method achieves **clear improvements on EMTD**, while on HDTF, the gains are competitive but less dominant. This is mainly because (1) our model is designed for upper-body/full-body talking avatars, whereas HDTF is a talking-head dataset with only cropped faces (hence we do not report HKC/HKV in Table 1); and (2) upper-body/full-body human video involves more challenging facial and hand details, where our Audio–Pose Prior Refocusing mechanism is particularly effective. The two datasets show that we **preserve strong talking-head performance while excelling in the more challenging upper-/full-body setting.**
> Regarding the FVD values in Tables 4 and 5, we have carefully rechecked and did not find incorrect bolding.
>
> ***
>
> 2. **More comparisons and inputs**
>
>
> We would like to further clarify that all demos shown in supplementary use **Audio + Reference Image + Text Description** as input.  Due to the 100MB limit on ICLR supplementary material, we could only include a subset of visual results. A more complete set of demos is provided at the link: [More Comparison Demo 1](https://anonymousdemo2026.github.io/anonymous-iclr26-demo/static/More\_Comparison1.mp4), [More Comparison Demo 2](https://anonymousdemo2026.github.io/anonymous-iclr26-demo/static/More\_Comparison2.mp4).
>
> ***
>
> 3. **Walking scenarios**
>
>
> Thank you very much for your suggestion. We have added a **walking-and-talking demo**, where the person moves while speaking. This can be found at the link:
> [Moving Demo](https://anonymousdemo2026.github.io/anonymous-iclr26-demo/static/moving.mp4), [Walking Demo](https://anonymousdemo2026.github.io/anonymous-iclr26-demo/static/walking.mp4).
> In these examples, the model maintains stable identity and natural motion even as the person moves across the scene.
>
> ***
>
> 4. **Citation of Suggested Work**
>
>
> We thank the reviewer for bringing this insightful work to our attention. DisFlowEm offers valuable perspectives on handling pose and expression flow. We have added a citation to the Related Work section in the revised manuscript.

---

> ### Author Response · Authors · 2025-11-27
>
> Dear Reviewer khjF,
>
> As the discussion period deadline approaches, we would like to kindly ask whether you have any thoughts on the rebuttal of our manuscript and whether it has influenced your assessment. We would be happy to address any remaining questions or concerns.
>
> Best regards,
> The Authors

---

### Comment · Area_Chair_oZNJ · 2025-11-25

Dear Reviewers,

A quick reminder that the authors have posted their responses. The discussion period ends on December 2, so please review the rebuttal and share any follow-up comments as soon as possible. Your timely input is greatly appreciated. Thanks.

Best,

Your ACs

---

### Author Response · Authors · 2025-12-01
**Summary for the AC (Part 3/3)**

### E. Reproducibility and Fairness (Addressing **DSoK**)
* **Training Details:** Clarifying the experimental setup for Reviewer **DSoK**, we detailed our two-stage training pipeline (Stage 1: Joint training; Stage 2: Balanced fine-tuning).
* **Fair Comparison:** Addressing Reviewer **DSoK**'s concern about the influence of text conditioning, we confirmed that identical text prompts were used for all competing methods to ensure a strictly fair comparison.
* **Reproducibility Commitment:** Responding to Reviewer **DSoK**'s call for open research, we have committed to releasing more details and providing interactive access to contribute to the community.

***

## Conclusion

We sincerely appreciate the valuable suggestions and positive recommendations from the reviewers. We believe that the aforementioned additions significantly enhance the completeness and value of the paper. The increased score from Reviewer **DSoK (4->6)** and the continued satisfaction of Reviewer **7rdx** reflect the effectiveness of our rebuttal.

With these improvements, we remain confident in our work's core contribution: bridging the gap between speech and motion to generate natural audio-driven avatars. By introducing a novel audio-pose prior and frame-wise interaction, we have effectively eliminated rigid movements—a design whose novelty and importance have been well-recognized by reviewers. We are confident that our work offers a solid contribution to the community.

Sincerely, Authors of Paper 5696

---

### Author Response · Authors · 2025-12-02
**Summary for the AC (Part 2/3)**

## 2. Addressed Concerns and Key Improvements

**Before the Data Leak**, we had already provided full responses to all questions. Notably, Reviewer **7rdx** expressed satisfaction with our answers and **kept a positive score of 8**, while **Reviewer DSoK raised the score from 4 to 6** after reviewing our clarifications and additional experiments. Although the remaining three reviewers did not have time to respond before the process was frozen, their concerns have been carefully addressed in our rebuttal:

***

### A. Enriched Visual Demos and Comparisons (Addressing **khjF**, **itA1**, **eXxq**, **DSoK**)
Reviewers pointed out that the supplementary material did not contain all the visual examples shown in the main paper. We would like to clarify that this was due solely to the **100MB ICLR supplementary file size limit**, not by choice. Many of our demos, especially those involving long speaking sequences and multi-method comparisons, could not fit within this constraint.

To address this concern, we now provide a significantly expanded set of qualitative results, covering all examples mentioned by the reviewers and many new comparisons:
* **More Baseline Comparisons:** As requested by Reviewer **DSoK**, we added extensive comparisons with recent audio-driven methods (including OmniAvatar), focusing on gesture naturalness and identity stability: [More Comparison Demo 1](https://anonymousdemo2026.github.io/anonymous-iclr26-demo/static/More_Comparison1.mp4) | [More Comparison Demo 2](https://anonymousdemo2026.github.io/anonymous-iclr26-demo/static/More_Comparison2.mp4) | [Qualitative Comparisons (Fig 4)](https://anonymousdemo2026.github.io/anonymous-iclr26-demo/static/Fig4_demo.mp4)

* **Diverse Motion Scenarios:** In response to Reviewer **khjF**’s concern about mostly static backgrounds, we demonstrated the model’s ability to handle walking, standing up, and large body movements while maintaining speech coherence: [Moving Demo](https://anonymousdemo2026.github.io/anonymous-iclr26-demo/static/moving.mp4) | [Walking Demo](https://anonymousdemo2026.github.io/anonymous-iclr26-demo/static/walking.mp4)

* **Multi-Person Generalization:** Addressing Reviewer **eXxq**’s question on generalization to multi-person scenes, we included a Multi-Person Demo result obtained via masked regional control: [Multi-Person Demo](https://anonymousdemo2026.github.io/anonymous-iclr26-demo/static/Multiperson_trial.mp4)

***

### B. Robustness and Ablation Studies (Addressing **itA1**, **eXxq**)
We clarified the reliability of our pose signals and the necessity of our interaction modules:
* **Pose Reliability:** To address Reviewer **itA1**’s worry that inaccurate pose priors might accumulate errors, we showed that our pose inputs are extracted using **Sapiens** (ECCV 2024 Best Paper Candidate), which provides dense and accurate keypoints (mouth, hands, pupils, etc.): [Pose Label Visualization](https://anonymousdemo2026.github.io/anonymous-iclr26-demo/static/pose_label.mp4)

* **Visual Ablation of Interaction & Refocusing:** Responding to **itA1**’s and **eXxq**’s requests for visual ablations, we provided demos showing that removing the Frame-Wise Audio–Video Interaction or the Audio–Pose Prior Refocusing results in less expressive motion and weaker alignment between audio and gestures. These ablations directly demonstrate that our modules are necessary for robust, natural motion: [Frame-wise Ablation Demo](https://anonymousdemo2026.github.io/anonymous-iclr26-demo/static/frame-wise_ablation.mp4) | [Refocusing Ablation Demo](https://anonymousdemo2026.github.io/anonymous-iclr26-demo/static/refocusing_ablation.mp4)

***

### C. Formal Presentation and Identity (Addressing **itA1**)
We took the feedback regarding the identity in Figure 1 seriously and planned to replace it with a formal example: [Formal Identity Demo](https://anonymousdemo2026.github.io/anonymous-iclr26-demo/static/formal_demo.mp4)

***

### D. Comprehensive Quantitative Evaluation (Addressing **eXxq**, **DSoK**)
* **OmniHuman Comparison:** OmniHuman 1.0 and 1.5 are closed-source and can only be accessed through an online interface, so evaluating all test samples requires running each case individually. Despite the substantial manual effort, we completed a full evaluation. To address Reviewer **eXxq**’s concern about missing closed-source baselines, we conducted a quantitative comparison with OmniHuman 1.0 and 1.5 on the EMTD dataset, referring to Response to Reviewer eXxq (Part 1/3). Our method achieves the best performance across all metrics.
* **Alignment Metrics:** Responding to Reviewer **eXxq**'s request for stronger perceptual metrics, we reported an F1-based Audio-Pose Alignment score (0.828). This significantly outperforms baselines (0.445–0.551), quantitatively proving the effectiveness of our model.

---

### Author Response · Authors · 2025-12-02
**Summary for the AC (Part 1/3)**

We sincerely appreciate the time and effort the reviewers and Area Chair have dedicated to assessing our work. We are encouraged by the positive consensus on the novelty of our approach and the recognized value of our research motivation.

Below, we summarize the key strengths acknowledged by the reviewers, the major improvements made during the rebuttal phase, and how we have addressed specific concerns with extensive experiments and visual demonstrations.

## 1. Consensus on Strengths: Novelty and Motivation

Reviewers have expressed strong recognition for the core contributions of our paper, particularly highlighting the novelty of our architecture and the practical significance of the problem we address:
* **Strong Motivation:** The reviewers agreed that our work addresses a critical gap in the field, specifically the issue of rigid movements and insufficient audio-pose interaction.
    * Reviewer **7rdx** stated that tackling rigid movements is a "practical problem" and our solution is "effective."
    * Reviewer **DSoK** praised the work for identifying a "critical yet often overlooked limitation" regarding independent modality modeling.
* **Novelty of ApoAvatar:** The proposed **Audio-Pose Prior Refocusing** and **Frame-Wise Audio-Video Interaction** mechanisms received broad acknowledgment from reviewers.
    * Reviewer **itA1** and **DSoK** found the refocusing mechanism "novel and interesting" and "well-designed."
    * Reviewer **eXxq** highlighted the "novel audio-prosody modulation" that effectively scales gesture intensity. Reviewer **eXxq** also praises the effective multimodal interaction ( Frame-Wise Cross-Attention) and the Flexible inference design.
    * Reviewer **khjF** noted that the interaction strategy is "interesting" for refining audio using video context.
* **Solid Evaluation and Presentation:** Reviewers **eXxq** and **7rdx** commended the "rigorous evaluation" and "extensive experiments" that demonstrate the framework's superiority. Reviewer **itA1** further noted that comparisons are "fair, thorough, and self-contained" and the presentation is "clear and well-structured". Additionally, Reviewer **eXxq** highlighted the "strong ablation support" that isolates specific module contributions.

---

### Meta-Review · Area_Chair_DmnM · 2026-01-10

**Summary:**

The paper proposes an Audio-Pose Prior Refocusing mechanism and a Frame-Wise Audio–Video Interaction module for audio-driven human video generation. Reviewers generally agree the ideas are interesting and well-motivated, and the experiments show some improvements in gesture expressiveness and audio-motion alignment. However, the improvements are often marginal, evaluations are limited in diversity and perceptual rigor, and key baselines or qualitative comparisons are incomplete. Concerns remain regarding reproducibility (closed-source backbone), insufficient discussion of text modality effects, limited generalization to multi-person or complex motion scenarios, and some methodological ambiguities. Overall, the contribution is incremental relative to existing work.

**Reviewer Concerns:**

The rebuttal addressed certain points, including providing additional demos, clarifying pose dropout and audio alignment, and citing missing references. Some weaknesses remain unresolved: reproducibility issues due to the closed-source Goku model, limited dataset diversity and generalization, inadequate discussion of text branch influence, and insufficient qualitative evaluation of hand/pose diversity and failure modes. While the authors clarified metrics and training, reviewers’ core concerns about robustness, fairness in comparisons, and completeness of evaluation still stand, leaving the overall impact of the proposed methods uncertain.

**Reviewer Scores:**

If all reviewers had participated fully in the discussion, the scores would likely remain mixed. Reviewers 1 and 3, who highlighted incremental improvements and dataset limitations, would probably keep marginally above-threshold ratings (6). Reviewers 2 and 5, focusing on methodological clarity, reproducibility, and experimental fairness, would likely remain below-threshold (4). Reviewer 4, already strongly positive, would maintain a high score (8), and Reviewer 5 indicated a willingness to raise their score to 6 after discussion. Despite some interest in the proposed mechanism, unresolved concerns regarding generalization, qualitative evaluation, and reproducibility collectively support a rejection. Moreover, the provided demos are only a few seconds long, raising doubts about the method’s ability to generate longer videos.

---

### Decision · Program_Chairs · 2026-01-26

Reject